# Functional nanoporous graphene superlattice

Hualiang Lv[1,2], Yuxing Yao [3,4], Mingyue Yuan[2], Guanyu Chen[2], Yuchao Wang[2,5], Longjun Rao[2], Shucong Li[3,6], Ufuoma I. Kara[1], Robert L. Dupont[1], Cheng Zhang [5] ✉, Boyuan Chen[1], Bo Liu [7], Xiaodi Zhou[1], Renbing Wu [2] ✉, Solomon Adera[8], Renchao Che [2] ✉, Xingcai Zhang [9] ✉ & Xiaoguang Wang [1,9,10] ✉

Two-dimensional (2D) superlattices, formed by stacking sublattices of 2D materials, have emerged as a powerful platform for tailoring and enhancing material properties beyond their intrinsic characteristics. However, conventional synthesis methods are limited to pristine 2D material sublattices, posing a significant practical challenge when it comes to stacking chemically modified sublattices. Here we report a chemical synthesis method that overcomes this challenge by creating a unique 2D graphene superlattice, stacking graphene sublattices with monodisperse, nanometer-sized, square-shaped pores and strategically doped elements at the pore edges. The resulting graphene superlattice exhibits remarkable correlations between quantum phases at both the electron and phonon levels, leading to diverse functionalities, such as electromagnetic shielding, energy harvesting, optoelectronics, and thermo-electrics. Overall, our findings not only provide chemical design principles for synthesizing and understanding functional 2D superlattices but also expand their enhanced functionality and extensive application potential compared to their pristine counterparts.

Two-dimensional (2D) superlattices, composed of stacked sublattices from 2D materials, have emerged as a robust platform for engineering materials with unique and tailored properties not presented in intrinsic materials[1,2]. The structural and compositional heterogeneities in 2D superlattices result in strongly correlated electronic and phonon phenomena, encompassing diverse quantum phases[3]. A notable example is the graphene Moiré superlattice, achieved by stacking graphene layers with precise control of twist angles or lattice misalignment degrees[4], leading to phenomena such as metal–insulator transition, ferromagnetism, anomalous quantum Hall effect, nematicity, and tunable Wigner crystal and Mott insulating states[5–8]. Current research on 2D superlattices predominantly focuses on the growth and stacking of pristine 2D material sublattices, by periodically stacking different sublattices or through misaligned stacking of identical sublattices, using techniques like molecular beam epitaxy, chemical vapor deposition, or atomic layer deposition[9–13]. However, the synthesis of

[1]William G. Lowrie Department of Chemical and Biomolecular Engineering, The Ohio State University, Columbus, OH 43210, USA. [2]Institution of Optoelectronic, Laboratory of Advanced Materials, Academy for Engineering & Technology, Department of Materials Science, Fudan University, Shanghai 200438, P. R. China. [3]Department of Chemistry and Chemical Biology, Harvard University, Cambridge, MA 02138, USA. [4]Division of Chemistry and Chemical Engineering, California Institute of Technology, Pasadena, CA 91125, USA. [5]Key Laboratory of Materials for High-Power Laser, Shanghai Institute of Optics and Fine Mechanics, Chinese Academy of Sciences, Shanghai 201800, P. R. China. [6]School of Engineering, Massachusetts Institute of Technology, Cambridge, MA 02139, USA. [7]College of Mechanical and Vehicle Engineering, Hunan University, Changsha 410082, P. R. China. [8]Department of Mechanical Engineering, University of Michigan, Ann Arbor, MI 48109, USA. [9]School of Engineering and Applied Sciences, Harvard University, Cambridge, MA 02138, USA. [10]Sustainability Institute, The Ohio State University, Columbus, OH 43210, USA. ✉e-mail: czhangseu@foxmail.com; rbwu@fudan.edu.cn; rcche@fudan.edu.cn; zhangxingcai@wteao.com; wang.12206@osu.edu

2D superlattice structures composed of chemically modified 2D material sublattices, such as those incorporating periodic vacancies or pores, presents a significant challenge[14]. In particular, the exploration of 2D superlattice structures in nanoporous graphene, a prominent member of chemically modified 2D materials with unique functionalities and potential applications[15,16], remains largely unexplored.

In this study, we present a chemical approach for the fabrication of a family of 2D superlattice structures by stacking graphene with monodisperse, nanometer-sized, square-shaped pores as sublattices to introduce additional degrees of freedom for tailoring the electronic and phonon structure of the superlattice. By employing in-situ growth and etching of cubic-shaped metal oxide nanoparticles on the graphene surface, we achieve a high density of monodisperse square nanopores. Compositional periodicity is also generated with selective doping of the pore edges with specific elements. Theoretical analysis indicates that the 2D superlattice structure induces a reconstruction of the electronic and phonon structures of graphene, leading to a flattened electronic band, the emergence of a phonon bandgap, and a significant electron–phonon coupling. As a result, the graphene superlattice exhibits unique properties, such as low-frequency dielectric relaxation, low thermal conductivity, low infrared emissivity, enhanced Seebeck effect, and photoluminescence effect, making it an attractive material for various applications including electromagnetic (EM) wave modulation, energy harvesting, thermal management, optoelectronics, and thermoelectrics. The synthesis strategies presented in this study offer a foundation for the development of

superlattices by stacking nanoporous 2D materials, paving the way for the design of 2D superlattices with tunable properties and unprecedented functionalities compared to pristine counterparts.

## Results

### Synthesis of porous graphene superlattices

We created graphene sublattices through the creation of a high density of monodisperse, square-shaped, nanometer-sized pores on graphene. The process involves three steps, as illustrated in Fig. 1a: (i) in-situ growth of cubic iron oxide nanoparticles ($Fe_3O_4$) template on the graphene surfaces (Supplementary Fig. 1); (ii) in-situ etching of the carbon atoms via the reduction reaction between graphene and $Fe_3O_4$ nanoparticles; and (iii) removal of $Fe_3O_4$ nanoparticle templates by hydrochloric acid. X-ray diffraction (XRD) confirmed the phase evolution of carbon upon reduction (Supplementary Fig. 2). Representative transmission electron micrographs in Fig. 1b demonstrate the resulting monolayer graphene possessing monodisperse, square nanopores with an average size of $7.1 \pm 0.8$ nm, same as the size of cubic $Fe_3O_4$ nanoparticles.

We expanded our method for stacking graphene sublattices to introduce nanopores on bilayer graphene, where the pores in one layer partially overlap with those in the other layer (Fig. 1c–e). The partially overlapped pores are ascribed to the presence of misaligned monodisperse $Fe_3O_4$ nanoparticles at both the upper and lower layers. This misaligned stacking of two porous graphene sublattices creates a distinct arrangement of carbon atoms with a periodicity of $3.2 \pm 0.7$ nm

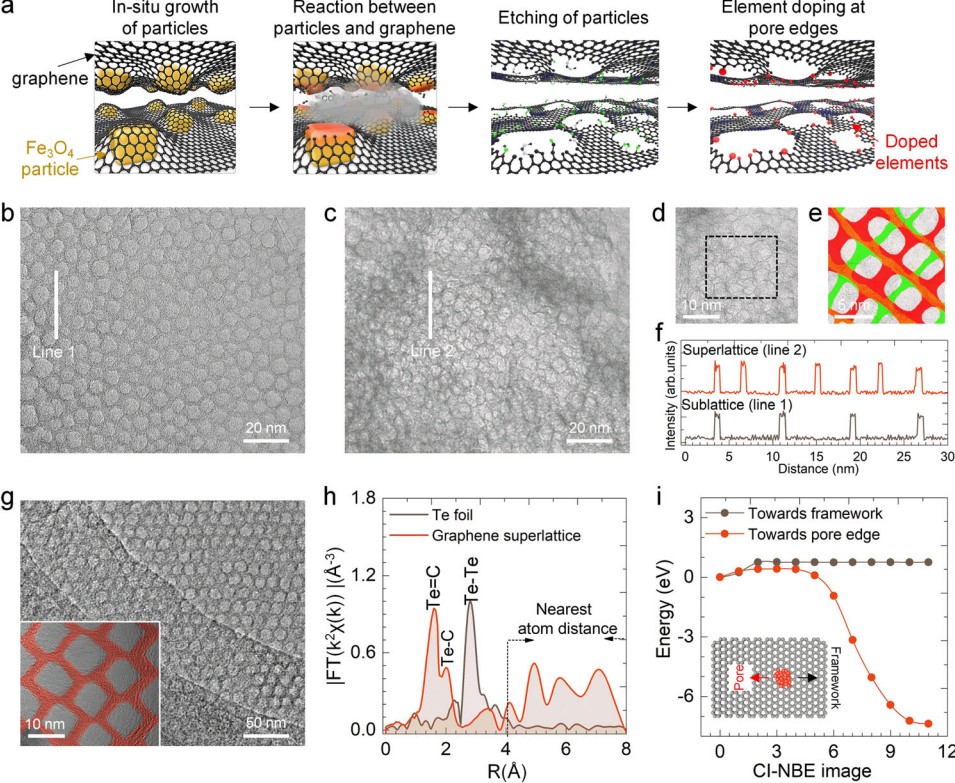

**Fig. 1 | Synthesis and structural characterization of element-doped porous graphene superlattice. a** Schematic representation of the synthesis process for graphene superlattice. The process to create element-doped graphene superlattice involves growing iron oxide nanoparticles on graphene, followed by carbon etching, nanoparticle removal with hydrochloric acid, and element doping of the graphene superlattice. **b–e** Transmission electron micrographs showing (**b**) monolayer graphene sublattice with monodisperse square-shaped nanopores and (**c–e**) bilayer graphene superlattice with partially overlapped square nanopore. The top and bottom layers in the graphene superlattice in (**e**) are

distinguished in red and green, respectively. **f** The periodic arrangement of carbon atoms in (**b**) monolayer graphene sublattice and (**c**) bilayer graphene superlattice. **g** Representative transmission electron micrograph of graphene with overlapped square nanopore. The inset in (**g**) presents a magnified transmission electron micrograph of graphene. **h** Fitted extended X-ray absorption fine structure (EXAFS) spectra of Te foil and Te-doped graphene superlattice. **i** Density functional theory (DFT) calculation of the energy of diffusion for doped Te atoms towards either a pore edge or a framework. The inset in (**i**) depicts the atomic diffusion path of doped element on the graphene surface.

(Fig. 1f). To examine the influence of misaligned sublattice stacking, we synthesized graphene with completely overlapped nanopores (Fig. 1g). High graphene flatness, a dense array of monodisperse nanometer-sized structures, and optimal carbon reduction reaction duration are crucial for creating ordered nanopores. Distortion or polydispersity in the nanopores is attributed to suboptimal conditions or irregular templates (Supplementary Figs. 3 and 4). Here we refer to bilayer graphene with partially overlapped square nanopores as graphene superlattice.

To achieve compositional heterogeneity, we precisely tune the type of functional groups at the pore edges through element doping. The pore edges of the obtained graphene superlattice possessed uncontrolled carbon (C)–oxygen (O) bonds, including –CO– and –COOH, as determined by Fourier transform infrared spectroscopy (FT-IR), X-ray photoelectron energy spectroscopy (XPS) and transmission electron microscopy (TEM) element mapping (Supplementary Fig. 5). To prepare hydrogen (H)-doped and O-doped nanoporous graphene, we annealed the graphene in the presence of a hydrogen gas atmosphere at 1200 °C and 800 °C, respectively, to transform all the edge bonds into –C–H and –C = O bonds (Supplementary Fig. 6). Further doping of a range of elements, including tellurium (Te), selenium (Se), nitrogen (N), phosphorus (P), sulfur (S), and boron (B), was achieved by a secondary thermal treatment to replace the –C–H bonds at the pore edges of H-doped porous graphene.

We employed Te-doped porous graphene superlattices as an example to analyze the state and distribution of doped elements on graphene surfaces. The extended X-ray absorption fine structure (EXAFS) analysis (Fig. 1h and Supplementary Fig. 7) showed two types of Te bonds located at the pore edges with lengths of 2.07 Å and 1.69 Å, in agreement with density functional theory (DFT) calculations (Supplementary Fig. 8). The appearance of consecutive peaks ranging from 3 Å to 8 Å, beyond the typical length of covalent or ionic bonds, suggests the formation of dangling Te bonds at the pore edges. DFT calculations using the climbing image nudged elastic band method revealed that Te atom diffusion towards the pore edge has a lower energy barrier (0.42 eV) and stronger bonding (−7.41 eV) compared to diffusion towards the framework, indicating preferential element doping at the pore edges (Fig. 1i, Supplementary Fig. 9 and Note 1).

## Electronic and phonon structure of graphene superlattice

We employed DFT calculations to compare the electronic structures of pristine bilayer graphene with graphene superlattices. Figure 2a illustrates that graphene superlattices exhibit a remarkable flattening of bands near the Fermi level, characterized by negligible band dispersion (less than 1 meV). This flat-band structure arises from the weak dispersion relation of kinetic energy resulting from the periodic electronic reconstruction caused by atomic stress between the overlapped and exposed carbon regions, leading to the emergence of a new $1s \rightarrow \sigma^*$ transition peak in X-ray absorption of the C K-edge (Supplementary Fig. 10 and Note 2). This electronic reconstruction leads to the formation of electron domain walls that confine Fermi electrons on either side, resulting in an equipotential Fermi surface and energy barriers with neighboring surfaces[17]. The electron trapping effect leads to a closer proximity of valence electrons in graphene superlattice to the Fermi level, resulting in the formation of multiple van Hoff singularities[18], as demonstrated by the distribution of electronic density of states in Fig. 2b.

Dopants in graphene superlattices result in an elevated electronic density of states on both sides of the Fermi level, a notable enhancement of electron–electron interaction, and an intensified spin–orbit coupling[19,20], leading to a weakening of the dispersion relation between electrons and kinetic energy, resulting in an electron system with

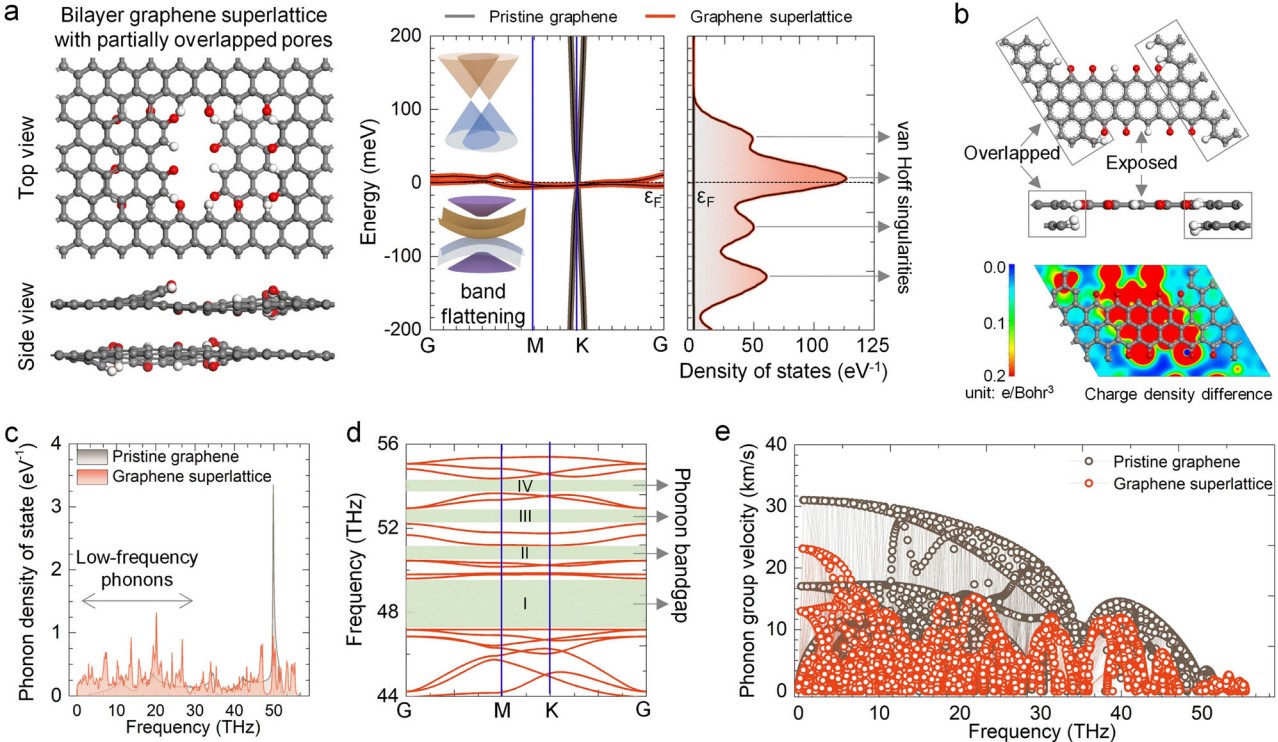

**Fig. 2 | Density functional theory (DFT) calculation of electronic and phonon structure of graphene superlattice. a** Band structure and projected density of states of pristine graphene and graphene superlattice. $\varepsilon_F$ represents the Fermi level. Arrows in (**a**) correspond to van Hoff singularities. **b** Charge density difference between the overlapped and exposed carbon atoms in graphene superlattice. **c–e** Phonon density of states, phonon band structure, and phonon group velocity of graphene superlattice. The overlap ratio of nanopores in graphene superlattice, defined by the ratio of the pore area covered by a neighboring graphene layer to the total pore area, was set to 0.5. The graphene superlattice was Te-doped. The green shaded areas, indicated by arrows in (**d**), represent the phonon bandgap.

infinite mass and an increased Coulomb interaction between electrons, thereby contributing to the enhanced band flattening. This computational analysis is supported by Hall carrier concentration measurements, revealing an approximate 400% increase in carrier concentration and a significant 99.8% decrease in carrier mobility in graphene superlattices compared to pristine graphene (Supplementary Fig. 11). Computational study reveals that electronic band flattening is influenced by various factors, such as structural periodicity, types and ratios of the dopants, and pore geometries (Supplementary Figs. 12–16 and Note 3). Bilayer graphene with partially overlapped circular nanopores or totally overlapped square nanopores lacks a flat-band structure due to the absence of a well-defined periodicity electron domain, attributing to the significant stress between the exposed and overlapped carbon atoms, which disrupts the formation of electron domain walls in these graphene structures.

We studied the effect of superlattice structure on the graphene phonon structure. Due to the harmonic oscillation of the atomic lattice[21], pristine graphene exhibits predominantly atom-propagated phonons with a frequency of 50 THz (Fig. 2c). In contrast, the periodic overlap of carbon atoms in bilayer graphene superlattices suppresses harmonic vibrations, leading to distinct phonon modes in the overlapped and exposed regions. The dopants at pore edges possess the same phonon vibrational modes that can give rise to clusters of new phonons[22,23], as evident by the vibrational density of states (Supplementary Fig. 17 and Note 4). The transmission of phonon clusters in graphene superlattices exhibits substantial coherent interference and elastic scattering[24], resulting in phonon cluster energy weakening and the emergence of a series of continuous standing waves in the low-frequency range of 10–30 THz, while the original high-frequency phonons disappear.

The absence of high-frequency phonons induces volume contraction of the first Brillouin zone, causing folding and compression of the phonon acoustic branches and resulting in phonon bandgaps in the high-frequency range[25], as compared to the pristine graphene (Fig. 2d and Supplementary Fig. 18). Calculations indicate that atomic participation ratios of graphene superlattices are 30–50% lower than those of pristine graphene (Supplementary Fig. 19), confirming that a decrease in the proportion of atoms participating in the same phonon mode directly leads to a reduction in phonon energy and cluster velocities (Fig. 2e). The resulting localization of phonons and the electron trapping effect leads to strong electron–phonon coupling, wherein electrons absorb phonons, undergo motion, and scatter to excite new phonons[26]. This continuous cycle gives rise to a prominent phonon drag effect, which significantly affects the trajectories and transport properties of electrons in graphene superlattice[27]. In summary, the graphene superlattice exhibits remarkable quantum phase correlations at both the electron and phonon levels, resulting in a diverse range of functionalities that surpass those observed in pristine graphene. Our synthesis strategies pave the way for the design of 2D materials[28,29], especially 2D superlattices with tunable properties and unprecedented functionalities compared to pristine counterparts.

## Electromagnetic wave modulation of graphene superlattice

The burgeoning advancements in wireless communications, particularly the fifth generation of mobile cellular network (5 G), have raised concerns over the underutilization of EM waves and disruptive signal interference[30,31]. Existing EM materials have limitations in their effectiveness, particularly with thicknesses of tens of micrometers. We investigate graphene superlattices to efficiently regulate EM waves and harness spatial EM energy. Figure 3a and Supplementary Fig. 20 demonstrate a remarkable dielectric dispersion behavior in the low-frequency 2–5 GHz region, which corresponds to the current wireless communication band, observed in the element-doped graphene superlattice. This distinctive dielectric dispersion arises from an electronic reconfiguration triggered by atomic stress at the intersections

of both overlapping and exposed carbon regions. This reconfiguration initiates the creation of electron domain walls, effectively capturing Fermi electrons on both sides and generating an interfacial potential difference between adjacent surfaces. When subjected to altered EM fields, this potential dynamically reorients in response to the modified electric field, giving rise to a behavior of polarization relaxation. This relaxation behavior is characterized by a sharp decrease in real permittivity ($\varepsilon'$) and a dielectric resonance peak in imaginary permittivity ($\varepsilon''$), resulting in a substantial increase in EM absorption, effectively dissipating the EM waves as joule heat and minimizing EM wave reflection and secondary interference[32]. We attribute this to the dielectric polarization relaxation, as demonstrated by the Cole–Cole curve (Supplementary Fig. 21 and Note 5), which arises from the periodic electron domain walls[33].

This dispersive behavior leads to an absorption efficiency of graphene superlattice in the 2–5 GHz region, within the thickness range of 5–100 μm, that surpasses that of pristine graphene by two orders of magnitude (Fig. 3b). Even at a thickness of only 50 μm, the absorption can exceed 60%. Due to its high EM absorption coefficient, the average EM shielding effectiveness of graphene superlattice can reach 99.99% (corresponding to an EM shielding effectiveness $SE_T > 40$ dB), effectively addressing secondary reflection-induced EM interference. The EM absorption capabilities of graphene superlattice in the micrometer thickness range surpass those of all existing EM materials (Supplementary Table 1). Direct comparison of EM shielding ability revealed that the EM shielding effectiveness and EM absorption efficiency per material thickness of graphene superlattice were two to four orders of magnitude higher than conventional 2D materials, including MXene and other chemically modified graphene, as shown in Fig. 3c. We demonstrated the ability to tune the EM absorption efficiency by adjusting the amounts of doped elements and shift the maximum EM absorption frequency by selecting the doped element type (Supplementary Fig. 22). This graphene superlattice selectively absorbs EM waves at a specific wavelength while allowing transmission of other bands, minimizing signal loss and offering potential integration into miniaturized circuits for efficient EM absorption.

The electron reconstruction triggered by the electronic domain not only impacts EM absorption but also notably reduces the relative complex permittivity ($\varepsilon_r$). This reduction significantly improves impedance matching, allowing incoming EM waves to penetrate the material more effectively, unlike the typical surface-skimming behavior observed with pristine graphene. In line with antenna design principles, our graphene superlattice film operates akin to an antenna, converting incoming waves into currents and generating a substantial electrical output. Figure 3d, along with Supplementary Figs. 23 and 24 and Note 6, elegantly illustrate the design and fabrication process of the graphene-superlattice device. As shown in Fig. 3e, the measured open circuit voltage ($U_{oc}$) of graphene superlattice increased from 0.1 V to 6.4 V with an increase in EM wave emission power from 1 mW to 10 W, respectively. The graphene superlattice-based device demonstrated a maximum power output of 0.31 W with a load resistance of 24 Ω when exposed to a 10 W source (Supplementary Fig. 25), and stable electricity output even after 1000 bending cycles or in different bending states (Supplementary Fig. 26). Furthermore, the overall efficiency of EM–DC conversion was calculated to be ~ 64% at an EM emission power of 100 mW, as shown in Fig. 3f. Notably, our device was able to efficiently harvest and convert EM waves from a 5 G cell phone, a wireless fidelity (Wi-Fi) router, a telecommunication tower, and a microwave to DC electricity that can be used to power electronic devices such as light-emitting diodes (LEDs) and digital hygro-thermometers, as shown in Fig. 3g and Supplementary Movies 1–4.

## Expanded applications of graphene superlattice

Beyond EM modulation, graphene superlattices unveil unique electronic and phonon characteristics, suggesting potential applications in

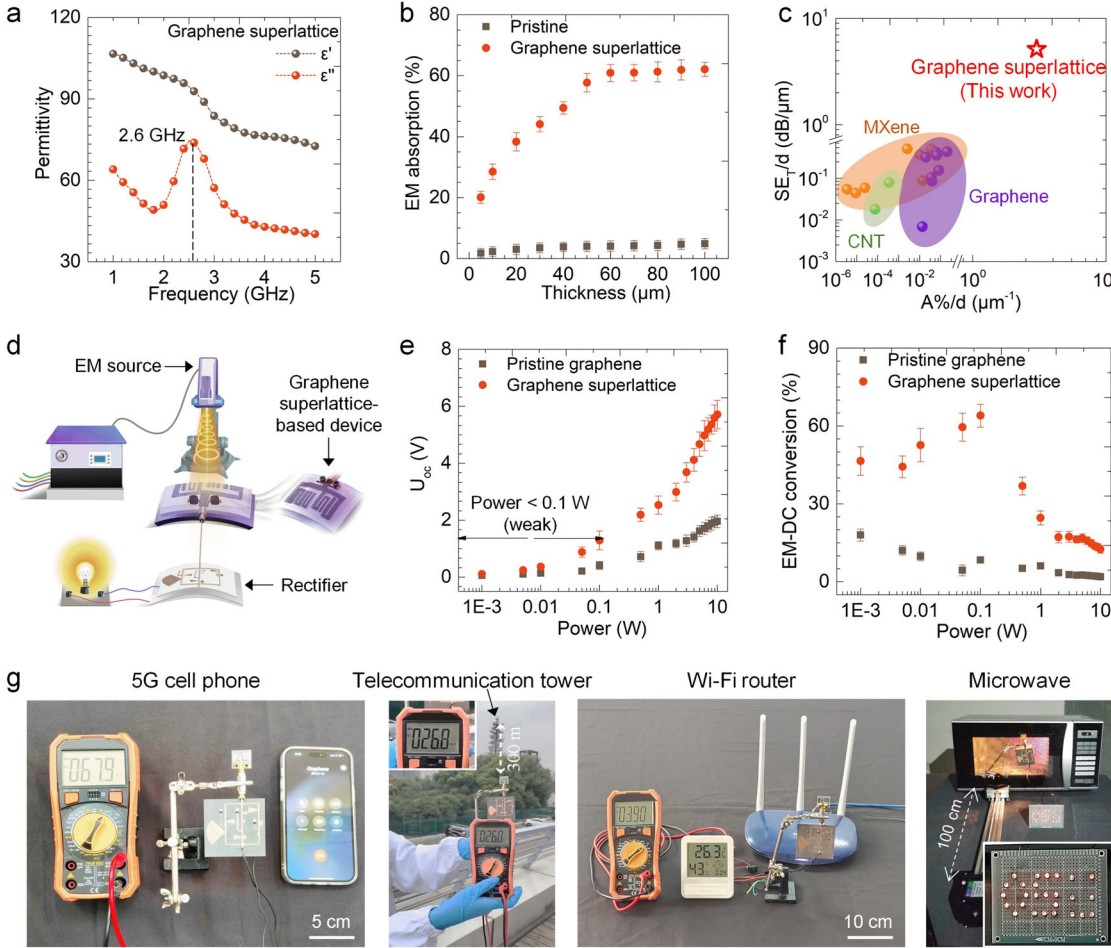

**Fig. 3 | Permittivity and electromagnetic wave modulation of graphene superlattice. a** Frequency-dependent permittivity of graphene superlattice. $\varepsilon'$ and $\varepsilon''$ represent the real and imaginary part of permittivity, respectively. The vertical dashed line marks the frequency corresponding to the dielectric loss peak. **b** Thickness-dependent electromagnetic (EM) wave absorption efficiency of pristine bilayer graphene and graphene superlattice. Error bars represent standard deviations from three independent measurements on the same physical samples. **c** Comparison of EM shielding effectiveness ($SE_T$) and EM absorption efficiency ($A\%$) averaged by EM absorbing material thickness of graphene superlattice with carbon nanotubes (CNTs) and other two-dimension (2D) materials. The data points compared in (**c**) are sourced from Supplementary References 82–96. **d** Experimental setup of graphene superlattice-based EM–electricity converter. **e** Open circuit

voltage ($U_{oc}$) and **f** EM–electicity conversion of pristine graphene and graphene superlattice-based EM–electricity converter as a function of EM emission power. The EM-to-electricity conversion efficiency is defined as the ratio of power generated by a graphene device, which converts input EM energy into direct current, to the input EM power. Error bars represent standard deviations from three independent measurements on the same physical samples. **g** Photographs of EM–electricity conversion by graphene superlattice-based EM–electricity converter from a 5 G cell phone, a Wi-Fi router, a telecommunication tower, and a microwave (Supplementary Movies 1–4). The output DC electricity can be used to power electronic devices such as a digital hygro-thermometer and a collection of LEDs arranged in an 'OSU' pattern. The graphene superlattice was Te-doped.

optics, electronics, and thermal sciences (see Supplementary Note 7 for detailed correlations). First, graphene superlattices exhibit multiple consecutive van Hoff singularities appear near the Fermi level (Fig. 2a). We developed a graphene superlattice-based smart EM wave window, which can selectively transmit, absorb, and shield EM waves by applying a voltage as low as 1.0 V (Fig. 4a). Second, in graphene superlattice, an increase in the energy level splitting and potential differences across the electron domain wall leads to the separation of electrons and holes, causing a remarkable photoluminescence effect within the visible light wavelength range (400–700 nm) in graphene superlattice, not being observed in pristine graphene (Fig. 4b). These findings highlight the high sensitivity of carrier transport in graphene superlattice with small bandgaps to external electric fields, making them promising candidates for applications in EM switch and optoelectronics.

Third, in the graphene superlattice, the presence of substantial coherent interference and phonon elastic scattering significantly diminishes in-plane lattice thermal conductivity, thereby leading to a

decrease in the overall thermal conductivity, as evidenced in Fig. 4c, Supplementary Fig. 27 and Note 8. This thermal conductivity, ranging between 6.4–4.4 W/mK, constitutes only 3.6–2.6% of the values observed in pristine graphene. This ultralow thermal conductivity not only provides desirable thermal insulation but also yields low infrared emissivity (~ 0.3 at the near-infrared band), making the graphene superlattice an ideal candidate for infrared stealth coatings (Supplementary Fig. 28). Furthermore, the combined effects of electron trapping and phonon drag significantly enhance the in-plane Seebeck coefficient of the graphene superlattice by 710–560% compared to pristine graphene in the temperature range of 300–500 K (Fig. 4d). Meanwhile, the electrical conductivity decreased by only 23.1–15.8% (Supplementary Fig. 29).

Current leading thermoelectric materials typically exhibit Seebeck coefficients within the 200–300 µV/K range, only around two to four times the Seebeck coefficient of our developed graphene superlattice. However, they possess lower thermal conductivities, typically around 1.0 W/mK, significantly less than our graphene superlattice. It is

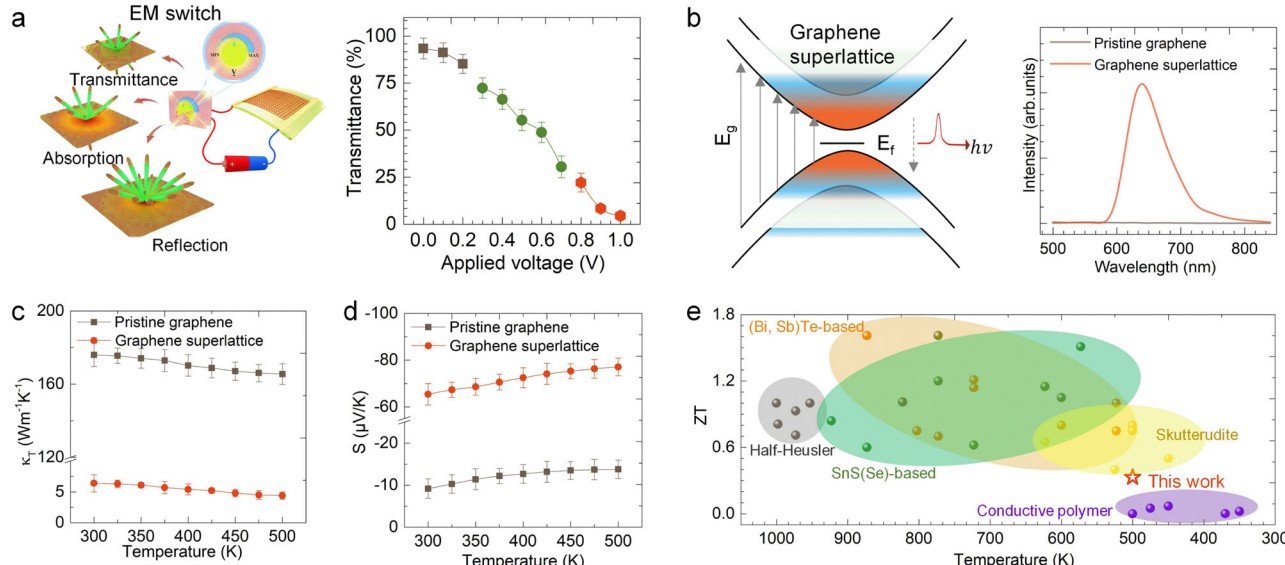

**Fig. 4 | Expanded applications of graphene superlattice. a** Tunable EM transmission of a graphene superlattice-based smart window via applied voltage. Error bars indicate standard deviations from three independent measurements on the same physical samples. **b** Photoluminescence intensity comparison between pristine graphene and graphene superlattice. $\varepsilon_F$ and $E_g$ represent the Fermi level and bandgap, respectively. $h\nu$ represents the energy of a photon. Upward arrows indicate the direction of electron transition, while the dashed downward arrow represents the direction of an electron transitioning from a higher energy orbital to a lower energy orbital. **c, d** Temperature-dependent in-plane thermal conductivity ($\kappa_T$) and in-plane Seebeck coefficient ($\alpha$) of pristine graphene and graphene superlattice. Error bars indicate standard deviations from three independent measurements on the same physical samples. **e** Comparison of in-plane figure of merit ($ZT$) values of graphene superlattice with state-of-the-art thermoelectric materials. The data points compared in (**e**) are sourced from Supplementary References 92–126. The graphene superlattice was Te-doped. The graphene superlattice achieved a maximum $ZT$ of 0.33, even at a low temperature of 500 K, surpassing pristine graphene by four orders of magnitude and demonstrating comparability to leading inorganic thermoelectric materials.

important to note that typical graphene materials differ significantly from leading thermoelectric materials in their Seebeck coefficient and thermal conductivity - the former often being two orders of magnitude lower and the latter three to four orders of magnitude higher. Yet, our graphene superlattice leads in electrical conductivity, surpassing traditional graphene by almost an order of magnitude. This substantial difference accentuates the thermoelectric performance of our graphene superlattice, resulting in remarkable in-plane figure-of-merit ($ZT$) values up to 0.33, even at a relatively modest temperature of 500 K (the effects on the out-of-plane parameters are detailed in Supplementary Figs. 29 and 30 and Note 9). This value is four orders of magnitude greater than pristine graphene, positioning our graphene superlattice among the most effective thermoelectric materials, as evidenced in Fig. 4e.

## Discussion

We have chemically synthesized a nanoporous bilayer graphene superlattice with partially overlapping pores, revealing remarkable correlations between electron and phonon quantum phases and enabling diverse applications beyond pristine graphene, including ultrathin EM absorption, EM energy harvesting and conversion, thermoelectric performance at low-to-intermediate temperatures, photoluminescence, and optoelectronic devices. Our study provides a chemical principle for the design and synthesis of a unique family of 2D superlattices, expanding our understanding of these structures and paving the way for the development of cutting-edge technologies. Future research efforts will focus on precise control of the lattice mismatching in nanoporous graphene sublattices to gain insights into their influence on the unique properties and functionalities of the superlattice. Our methodology can be extended to the creation of 2D superlattices using other conventional 2D materials, further broadening the range of materials including 2D and porous materials[15,16,34–40] with tailored properties and diverse applications.

## Methods

### Materials

Iron acetylacetonate, oleic acid, oleyl amine, ammonia (28 wt % in water), hydrogen peroxide (30 wt % in water), sodium chloride, sodium molybdate dehydrate, thiourea, ferrous chloride, cobalt acetate, ethylene glycol, zinc acetate, lanthanum oxide, cobalt nitrate, tetraethyl orthosilicate, sodium hydroxide, oxalic acid, strontium carbonate, cobalt oxalates, bismuth nitrate pentahydrate, diboron trioxide, 1-butyl-3-methlyimidazolium hexafluorophosphate (BmimPF$_6$), silver nitrate, stannous chloride, sulfur powder, ethylenediaminetetraacetic acid disodium, copper nitrate, hexadecyltrimethylammonium bromide, formaldehyde, mercury oxide, mercury, cyclohexane, isopropanol, iron pentacarbonyl, 1,3-dimethylimidazoline-2-selenone, hydrochloric acid (38 wt % in water), tellurium power, selenium powder, potassium hydroxide, sulfuric acid and sulfide powder were purchased from Sigma-Aldrich. Raw graphene and multiwall carbon nanotubes were obtained from XF-NANO Tech. Co. All chemical reagents are analytical pure reagents and were used without further purification. Polydimethylsiloxane (PDMS) film was purchased from Shanghai Muke Technol. Co. Deionized water used in all the experiments was purified using a Simplicity C9210 Milli-Q water purification system.

### Synthesis of graphene superlattice

Initially, a mixture of iron acetylacetonate (1.4 g), raw bilayer graphene (10 mg) synthesized via chemical vapor deposition (CVD), oleic acid (10 mL), and oleyl amine (10 mL) was prepared and stirred for 30 min. The mixture was then heated under a nitrogen flow at a temperature range of 110–120 °C with vigorous stirring for 2 h. Subsequently, the temperature was increased to 220 °C at a heating rate of 5 °C/min and held for 30 min, followed by heating to 300 °C for an additional 30 min at a rate of 2 °C/min. After cooling to room temperature, the graphene coated with cubic Fe$_3$O$_4$ nanoparticles was collected by centrifugation

after dispersion in cyclohexane and isopropanol. The obtained graphene coated with cubic $Fe_3O_4$ nanoparticles was subjected to a heat treatment at 300 °C for 60 min under a continuous nitrogen flow to remove organic solvents. Subsequently, the temperature was gradually increased to 650 °C at a rate of 5 °C/min and maintained for 0.5 h to induce the formation of square-shaped nanometer-sized pores on the bilayer graphene surface. After cooling, the graphene was dispersed in a hydrochloric acid solution with a pH of 3–4 to remove the coated $Fe_3O_4$ nanoparticles, resulting in the formation of squared nanopores. By employing raw graphene with different layer numbers, porous graphene with various layer numbers can be fabricated.

### Synthesis of graphene with completed overlapped nanopores

We produced graphene with fully overlapped nanopores using a bottom-up approach involving the thermal decomposition of oleic acid iron-coated sodium chloride (NaCl) particles. Initially, 0.1 g of NaCl particles, ranging from 5 to 10 μm, were submerged in 10 mL of oleic acid iron solution. This step ensures the adsorption of the oleic acid iron onto the NaCl particles, resulting a coated layer of oleic acid iron on the NaCl surface. During thermal decomposition at 500 °C with a nitrogen gas flow, the iron oleate on the NaCl surface undergoes a transformation into cubic $Fe_3O_4$ nanoparticles. Simultaneously, carbon sheets form, encapsulating these cubic nanoparticles and incorporating the $Fe_3O_4$ into carbon nanosheets. Subsequent to the thermal decomposition, the composite material is immersed in water to dissolve and remove the NaCl substrate. An acid wash follows, selectively extracting the $Fe_3O_4$, resulting in the production of aligned, porous carbon nanosheets. Lastly, graphene with completely overlapped nanopores was obtained by drying in a vacuum oven for 12 hours, followed by thermal reduction at 1500 °C for 2 hours in a hydrogen gas flow.

### Selective element doping at pore edges of graphene superlattice

To achieve selective doping of elements at the pore edges of the graphene superlattice, a stepwise annealing process was employed. The graphene superlattice was subjected to annealing at 600 °C, 800 °C, and 1200 °C for 1 h to progressively decompose the –OH, –CO, and –COOH groups at the pore edges, respectively. The first two annealing steps were carried out under a nitrogen gas flow, while the final step was conducted in a mixture gas consisting of 5 vol % hydrogen and 95 vol % nitrogen. Selective doping of O and H at the pore edges of the bilayer porous graphene was achieved after the second (800 °C) and third step (1200 °C) of annealing, respectively. Furthermore, the selective doping of other elements such as B, N, P, S, Te, and Se at the pore edges of graphene was achieved through thermal decomposition at various temperatures (Supplementary Table 2).

### Structural characterization of porous graphene

Graphene superlattice phase identification was conducted using a Bruker D8 ADVANCE XRD equipped with Cu Kα radiation (wavelength of 0.15406 nm). The morphology of the nanoparticles and porous graphene was investigated using a Joel JEM 2100 F transmission electron microscope (TEM) with a 200 kV field emission. X-ray photoelectron spectroscopy (XPS) analyses were carried out using an Escalab 250Xi spectrometer. The graphitization quality was evaluated using a Jobin Yvon HR 800 confocal Raman spectroscope. Synchrotron X-ray absorption near-edge structure (XANES) analysis was performed at Beamline 33BM-C in the XOR Division of the Advanced Photon Sources. Extended X-ray absorption measurements were conducted at beamline 7-BM of the National Synchrotron Light Source II, Brookhaven National Laboratory. Chemical bonding analysis of graphene was carried out using a Nicolet iS50 Fourier Transform Infrared Spectrometer (FT-IR). Photoluminescence performance was recorded at room temperature using a LifeSpec II fluorescence lifetime spectrometer

equipped with a diode laser emitting at a wavelength of 365 nm. Infrared emissivity was measured using a PerkinElmer Spotlight 200i FT-IR spectrometer along with an IR integrating sphere.

### Evaluation of electromagnetic shielding and absorption of graphene superlattice

The EM frequency-dependent scattering parameters (i.e., $S_{11}$ and $S_{21}$) and permittivity of graphene were measured via the single coaxial-line method using an Agilent PNA N5224A vector network analyzer. The corresponding absorption efficiency (A%) and shielding effectiveness ($SE_T$) can be calculated as[41,42]:

$$A\% = (1 - |S_{11}|^2 - |S_{21}|^2) \times 100\% \qquad (1)$$

$$SE_T = -10\log|S_{21}|^2 \qquad (2)$$

We note here that the uncertainty in the permittivity and scattering parameter measurements is about 10–15 % due to variations originating from the smoothness of sample surfaces.

### Characterization of in-plane thermal conductivity, electric conductivity and Seebeck coefficient of graphene superlattice

To obtain the in-plane Seebeck coefficient and electrical conductivity, a precise amount of bilayer graphene powder was dispersed in an ethanol solution. A film approximately $0.20 \pm 0.02$ mm thick was created using vacuum filtration. This filtered film was shaped into strips measuring 6 mm × 4 mm in length and width. Around 15 of these strips were oriented and stacked to form a film with 3 mm thick. The Seebeck coefficient and electrical conductivity were evaluated along the length of the film using the ULVAC-RIKO ZEM-3 apparatus in a helium atmosphere at a heating rate of 5 °C/min.

To assess the in-plane thermal conductivity, the filtered film was sectioned into strips measuring 15 mm in length and 5 mm in width. Approximately fifty of these strips were stacked to create a film approximately 10 mm thick. Heat conduction was directed along the length of the stacked film during the evaluation using the Netzsch HFM446 heat flow thermal instrument, ensuring precise measurement of in-plane thermal conductivity. Based on the in-plane Seebeck coefficient, electrical conductivity, and thermal conductivity, figure of merit (ZT) was calculated as[43]:

$$ZT = \sigma \alpha^2 T / \kappa_T \qquad (3)$$

where $\sigma$ represents electrical conductivity, $\alpha$ denotes the Seebeck coefficient, $\kappa_T$ corresponds to thermal conductivity, and $T$ is the temperature.

## Data availability

Relevant data supporting the key findings of this study are available within the article and the Supplementary Information file. All raw data generated during the current study are available from the corresponding authors upon request.

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

## Acknowledgements

X.W. would like to acknowledge the financial support from the startup fund of The Ohio State University (OSU), OSU Sustainability Institute Seed Grant, OSU Materials Research Seed Grant Program, funded by the Center for Emergent Materials, an NSF-MRSEC, grant DMR-2011876, the Center for Exploration of Novel Complex Materials, and the Institute for Materials Research.

## Author contributions

H.L., R.W., C. Z.; R.C., S.A., M.Y., X. Zhang, and X.W. conceived and designed the experiments, supervised the research, and contributed to manuscript writing. H.L., M.Y., Y.Y., S.L., R.L.D., U.I.K., B.C., C.Z., X.Zhou, Y.W., and R.W. conducted material synthesis and performed various characterizations. C.Z., Y.W. and L.R. designed the graphene device and measured its EM–electricity output. H.L., L.R., B.C., and X.Zhou conducted permittivity, thermal conductivity, Hall effect, and Seebeck characterizations. H.L and B.L. performed molecular dynamics simulations. G.C and B.L. carried out first-principles density functional theory simulation and modular dynamic analysis. All authors contributed to data interpretation, discussions, and manuscript preparation. H.L., Y.Y., M.Y., G.C., and Y.W. contributed equally to this work.

## Competing interests

The authors declare no competing interests.
