## [Peer Review File · Nature Communications]

REVIEWER COMMENTS

Reviewer #1 (Remarks to the Author):

The authors reported an interesting approach for creating unique graphene superlattices by incorporating monodisperse nanopores and doped elements at the pore edges. The demonstrated method is interesting and could be applied to other 2D material systems. However, the manuscript needs major revision to fully address the following comments before it can be considered for publication in Nature Comms.

1. Why is the Fe₃O₄ nanoparticle chosen as the template? What are the fundamental and technical reasons/benefits? Could nanoparticles of other materials be used for the same purpose?
2. High-resolution TEM and STEM analysis would better reveal the structural features for the obtained superlattices. The TEM images shown in current Fig. 1 do not provide too much useful information regarding the structures, arrangement, and lattice information.
3. The authors highlighted that the graphene superlattices could achieve versatile quantum phase correlations at electron and phonon levels. However, it is unclear (and to some extent overstressing) how the simulation results in Fig. 2, the EM results in Fig. 3, and the several applications in Fig. 4 are related to such claims.
4. What quantum phases and what possible correlations at the electron/phonon levels can be obtained and how they contribute to/benefit the applications in Figs. 3 and 4 (as well as in supplementary figures)?
5. How do the superlattice structures impact the characterized properties and related applications shown in this paper?

Reviewer #2 (Remarks to the Author):

The authors have used decomposition and annealing method to generate nanopores in graphene layers and doped with guest atoms to endow graphene with multiple functions. However, before further consideration, the following concerns are suggested to be resolved.

- 1) Bilayer graphene cannot be called as "superlattice", is there any more proof? As superlattice materials are multilayer films grown alternately by two different components in thin layers ranging from a few nanometers to tens of nanometers and maintaining strict periodicity. We need to see the periodicity in the sectional dimension. Cs-TEM is needed to get precise information.
- 2) From the measurement part, thermal conductivity seems to be measured in the out-plane direction. It is not in line with the electrical properties. The measured thermal conductivity in the out-plane direction after hot-pression will be much lowered, causing too high ZT, but not true. Please check.
- 3) As graphene is anisotropic, to make the conclusion convincing, both the in-plane and out-plane

thermoelectric properties should be measured. It is supposed to be quite different. The effects of nanopores and Te-doping on the thermoelectric properties in different directions should be discussed in detail.

4) Wondering how the electric conductivity and Seebeck coefficient of the ultra-thin graphene was measured on ZEM-3. And what is the direction? In-plane or out-plane?

5) For Fig3b, is there an upper limit of the EM ab.%, as it was increased with thickness. What is the density of the superlattice films? How about the curve of EM ab.% vs. density? "graphene" with thickness of 50 μm cannot be called graphene, it is graphitic.

6) The thermal conductivity is incredibly low, please give information of the electrical conductivity and lattice thermal conductivity to make it convincing.

Reviewer #3 (Remarks to the Author):

Major revisions.

Currently, it is a great challenge for fabrication of 2D superlattice structures composed of chemically modified 2D material sublattices, especially the exploration of 2D superlattice structures in nanoporous graphene. In this work, the authors reported a chemical method to synthesize a family of 2D superlattice structures to tailor the electronic and phonon structure of superlattice, through introducing additional degrees of freedom. This is an interesting and novel work that deserves publication in Nature Communications. However, there are still a couple of issues needed to note and it may be published after major revision. The comments are as follows:

1. What is the advantage of graphene super-lattice for the ZT factor? Especially, many materials have the ZT factors larger than 1.6 such as Lidong Zhao et al. published papers in Science. The authors did not add these data in the figure.

2. I cannot see the better performance of thermoelectricity for the reported graphene superlattice including the various thermoelectric factors such as Seebeck coefficient and so on. They need to compare most of the best materials currently published online.

3. Please check all the full names of the abbreviations given when they first appeared. Besides, all the full names should be uniform. There are numerous lines in most figures, when jointing the images. Turn off borders on images, and remove the redundant lines.

4. In the Introduction part, when reviewing the background of 2D superlattices, more classic references should be cited, instead of citing papers published in the last two years.

Reviewer #1:

The authors reported an interesting approach for creating unique graphene superlattices by incorporating monodisperse nanopores and doped elements at the pore edges. The demonstrated method is interesting and could be applied to other 2D material systems. However, the manuscript needs major revision to fully address the following comments before it can be considered for publication in Nature Comms.

Response: We thank the reviewer for their careful reading of our manuscript and for noting that our approach is interesting for creating unique graphene superlattices by incorporating monodisperse nanopores and doped elements at the pore edges.

1. Why is the Fe_3O_4 nanoparticle chosen as the template? What are the fundamental and technical reasons/benefits? Could nanoparticles of other materials be used for the same purpose?

Response: We wish to clarify that our objective in creating a graphene superlattice relies on establishing a high-density array of monodisperse, nanometer-sized square-shaped pores on bilayer graphene. The process encompasses three crucial steps:

- (i) In-situ growth of cubic transitional oxide nanoparticle templates on graphene surfaces.
- (ii) In-situ etching of carbon atoms via the reduction reaction between graphene and the nanoparticle templates.
- (iii) Removal of the reduced templates using hydrochloric acid.

The selection of Fe_3O_4 nanoparticles as templates is justified by several advantages:

- (i) Defined structure: The size and square shape of Fe_3O_4 can be precisely adjusted using the classic thermal decomposition method, a technique that has significantly advanced in recent years (*Adv. Mater.* 2010, 22:2729; *Chem. Soc. Rev.* 2009, 38:2532).
- (ii) Reactivity with carbon: Fe_3O_4 interacts with carbon at specific annealing temperatures, facilitating the removal of carbon atoms and consequentially forming the desired square-shaped nanopores.
- (iii) Ease of removal: The post-reaction Fe_3O_4 template can be readily dissolved using hydrochloric acid.

Fe_3O_4 emerges as a suitable choice for crafting nanopore templates on graphene. Although other oxides like CoO , Fe_2O_3 , Co_3O_4 , and CoFe_2O_4 could serve a similar purpose if aligned with the aforementioned criteria, the primary challenge is precisely synthesizing these oxides into uniform, nanometer-sized, square-shaped nanoparticles using the thermal decomposition method. For instance, replacing the entire amount of iron acetylacetonate $[\text{Fe}(\text{acac})_3]$ with equimolar quantities of cobalt acetylacetonate $[\text{Co}(\text{acac})_2]$ and nickel acetylacetonate $[\text{Ni}(\text{acac})_2]$ resulted in the formation of CoO and NiO nanoparticles, respectively. Unfortunately, we observed irregular ellipsoidal and polyhedral formations in both CoO and NiO , deviating significantly from our intended uniform structure. NiO particularly showed notable agglomeration, hindering a consistent and well-dispersed distribution. These structural discrepancies and irregularities may stem from differing nucleation and growth tendencies of Co and Fe ions, suboptimal ratios of oleic acid to oleylamine, and potential variations in reaction temperature.

While the potential of other oxides remains theoretically feasible, substantial efforts are crucial to refine their synthesis and achieve the desired structures.

To address the reviewer's comment, we have added the following text to the revised manuscript.

“High graphene flatness, a dense array of monodisperse nanometer-sized structures, and optimal carbon reduction reaction duration are crucial for creating ordered nanopores. Distortion or polydispersity in the nanopores is attributed to suboptimal conditions or irregularly templates (Supplementary Figs. 3 and 4).”

In addition, we have added Supplementary Fig. 4 to the revised Supplementary Materials:

Supplementary Fig. 4 – Morphology of CoO and NiO nanoparticles. a, b, TEM images revealing irregular ellipsoidal and polyhedral (a) CoO and (b) NiO particles, formed through complete substitution of $\text{Fe}(\text{acac})_3$ with $\text{Co}(\text{acac})_2$ and $\text{Ni}(\text{acac})_2$, respectively.”

2. High-resolution TEM and STEM analysis would better reveal the structural features for the obtained superlattices. The TEM images shown in current Fig. 1 do not provide too much useful information regarding the structures, arrangement, and lattice information.

Response: We wish to clarify that our work centers on creating graphene superlattices by systematically layering monolayer graphene sublattices, each featuring evenly distributed square nanopores at the nanometer scale. The key attribute of our superlattice design lies in the alignment of nanopores between the top and bottom layers of bilayer graphene, resulting in a unique periodic structure. When examining our superlattice, the most effective approach is utilizing the low-resolution TEM mode. This method offers an excellent top-down view of the nanopore arrangement in both layers, showcasing the precise dimensions and shapes within the graphene structure. Our variant of graphene superlattice stands apart from mainstream 2D superlattices formed by rotating a pristine monolayer of 2D material at specific angles, often known as ‘magic angle graphene’. These structures usually display periodic Moiré lattice patterns, a key indication of such superlattices, which necessitates confirmation through HRTEM-STEM imaging. Furthermore, our reported graphene superlattice notably differs from traditional 2D superlattices, recognized as 2D superlattice heterostructures oriented along the z-axis, requiring cross-sectional HAADF-STEM imaging to depict the periodic stacking.

3. The authors highlighted that the graphene superlattices could achieve versatile quantum phase

correlations at electron and phonon levels. However, it is unclear (and to some extent overstating) how the simulation results in Fig. 2, the EM results in Fig. 3, and the several applications in Fig. 4 are related to such claims.

Response: In our study, we explore the significant correlation between the electronic structure, phonon structure, and their interaction in the novel graphene superlattices. Each aspect contributes to the material's multifaceted applications, including EM modulation, luminescent properties, low infrared emissivity, and thermoelectric characteristics. To address the reviewer's comment, we provide a comprehensive understanding of the intricate connections among these facets:

(i) *Connection of electron structure with EM modulation and luminescence performance:*

Our simulations demonstrate a distinctive band flattening in the graphene superlattices we developed, presenting an almost imperceptible band dispersion (below 1 meV) near the Fermi level. This flat-band structure emerges from the weak dispersion of kinetic energy due to periodic electronic reconstruction caused by atomic stress between overlapped and exposed carbon regions. This reconstruction forms electron domain walls, trapping Fermi electrons on both sides, generating an electronic trap effect. As a result, the electronic domain undergoes reconstruction, introducing an interfacial electronic potential difference between neighboring surfaces. Under altered EM fields, this potential dynamically reorients in line with the modified electric field, resulting in a polarization relaxation. This behavior contributes to the absorption of EM energy, a phenomenon uncommon in pristine graphene (*Prog. Mater. Sci.* 2022, 127:100946; *Adv. Mater.* 2022, 34:2107538).

The electron reconstruction caused by the electronic domain results in a distinctive permittivity dispersive behavior, notably reducing the relative complex permittivity (ϵ_r). This reduced permittivity allows enhanced impedance matching, enabling incoming EM waves to penetrate the interior more effectively than the surface, as observed with pristine graphene. Inspired by antenna design principles, we have engineered the graphene superlattice film to act as an antenna, converting incoming waves into currents and producing a substantial electrical output (Fig. 3d-g of the revised manuscript).

Observations of the graphene superlattice's band indicate multiple van Hoff singularities near the Fermi level (Fig. 2a of the revised manuscript). These arise from electron domains, splitting the Fermi surface and creating several new Fermi surfaces with distinct energy differences. This capability allows the superlattice to use an external electric field to overcome potential differences, exciting its Fermi electrons and achieving enhanced conductivity, linearly influencing the permittivity. This linear increase in permittivity positions the film for various stages of EM transmission, absorption, and shielding as the permittivity progressively rises, indicating EM switch ability (Fig. 4a of the revised manuscript). The increased energy level splitting and potential differences across the electron domain wall result in the separation of electrons and holes, causing a remarkable photoluminescence effect within the visible light wavelength range (400–700 nm) unique to the graphene superlattice, not observed in pristine graphene (Fig. 4b of the revised manuscript).

(ii) *Connection of phonon structure with reduced thermal conductivity:*

Examining the superlattice structure's impact on the graphene phonon structure, pristine graphene predominantly exhibits atom-propagated phonons with a frequency of 50 THz (Fig. 2c of the revised manuscript). In contrast, the periodic overlap of carbon atoms in bilayer graphene superlattices suppresses harmonic vibrations, resulting in distinct phonon modes in the overlapped and exposed regions. The transmission of phonon clusters in graphene superlattices introduces substantial coherent interference and elastic scattering, weakening the energy of the phonon clusters and creating a series of continuous standing waves in the low-frequency range of 10–30 THz. Consequently, the phonon mean free path reduces significantly, leading to pronounced localization and, consequently, a decline in thermal conductivity and thermal infrared emission (Fig. 4c of the revised manuscript and Supplementary Fig. 33 of the revised Supplementary Materials).

(iii) *Connection of electronic-phonon interaction with thermoelectric behavior:*

The localization of low-frequency phonons and the electron trapping effect induce strong electron–phonon coupling, with electrons absorbing phonons and scattering to excite new phonons. This ongoing cycle gives rise to a pronounced phonon drag effect, significantly affecting the trajectories and transport properties of electrons in the graphene superlattice. This phonon drag effect positively impacts the Seebeck coefficient (Fig. 4d of the revised manuscript). The increased Seebeck coefficient, combined with reduced thermal conductivity, results in a nearly two orders of magnitude higher ZT value compared to pristine graphene, highlighting its exceptional potential for thermoelectric applications (Fig. 4f of the revised manuscript).

To address the reviewer's comment, we have added the following texts to the revised manuscript:

“Fig. 3a and Supplementary Fig. 24 demonstrate a remarkable dielectric dispersion behavior in the low-frequency 2–5 GHz region, which corresponds to the current wireless communication band, observed in the element-doped graphene superlattice. This distinctive dielectric dispersion arises from an electronic reconfiguration triggered by atomic stress at the intersections of both overlapping and exposed carbon regions. This reconfiguration initiates the creation of electron domain walls, effectively capturing Fermi electrons on both sides and generating an interfacial potential difference between adjacent surfaces. When subjected to altered EM fields, this potential dynamically reorients in response to the modified electric field, giving rise to a behavior of polarization relaxation.”

“The electron reconstruction triggered by the electronic domain not only impacts EM absorption but also notably reduces the relative complex permittivity (ϵ_r). This reduction significantly improves impedance matching, allowing incoming EM waves to penetrate the material more effectively, unlike the typical surface-skimming behavior observed with pristine graphene. In line with antenna design principles, our graphene superlattice film operates akin to an antenna, converting incoming waves into currents and generating a substantial electrical output.”

“Beyond EM modulation, graphene superlattices unveil unique electronic and phonon characteristics, suggesting potential applications in optics, electronics, and thermal sciences (see Supplementary Note 8 for detailed correlations).”

In addition, we have added Supplementary Note 8 to the revised Supplementary Materials:

“Supplementary Note 8: The correlation between electron and phonon structures in extended applications

(i) Connection of electron structure with electromagnetic modulation and luminescence performance:

Observations of the graphene superlattice’s band indicate multiple van Hoff singularities near the Fermi level (Fig. 2a). These singularities stem from electron domains that split the Fermi surface, creating new Fermi surfaces with differing energy levels. This property enables the graphene superlattice to harness external electric fields to excite Fermi electrons, consequently enhancing conductivity and linearly influencing the permittivity. The linear increase in permittivity indicates the film’s suitability for EM transmission, absorption, and shielding as it progressively rises, suggesting EM switchability (Fig. 4a). The increased energy level splitting and potential differences across the electron domain wall lead to the separation of electrons and holes, producing a notable photoluminescence effect in the visible light wavelength range (400–700 nm), distinct to the graphene superlattice and not observed in pristine graphene (Fig. 4b).

(ii) Connection of phonon structure with reduced thermal conductivity:

Upon examining the impact of the superlattice structure on the graphene phonon structure, it becomes apparent that pristine graphene predominantly features atom-propagated phonons at a frequency of 50 THz (Fig. 2c). However, in bilayer graphene superlattices, the periodic overlap of carbon atoms hinders harmonic vibrations, creating distinct phonon modes in the overlapped and exposed regions. This unique transmission of phonon clusters results in significant coherent interference and elastic scattering, weakening the energy of the phonon clusters. This generates a series of continuous standing waves in the low-frequency range of 10–30 THz, substantially reducing the phonon mean free path. Consequently, this reduction leads to notable localization and a decline in thermal infrared emission (Fig. 4c and Supplementary Fig. 33).

(iii) Connection of electron–phonon Interaction with thermoelectric behavior:

The localization of low-frequency phonons and the electron trapping effect causes a strong electron–phonon coupling. As a result, electrons absorb phonons and scatter to excite new phonons, initiating a pronounced phonon drag effect. This effect significantly impacts the trajectories and transport properties of electrons within the graphene superlattice. This substantial phonon drag positively influences the Seebeck coefficient (Fig. 4d). The amplified Seebeck coefficient, combined with reduced thermal conductivity, results in a nearly two orders of magnitude higher ZT value compared to pristine graphene, showcasing its exceptional promise for thermoelectric applications (Fig. 4f).”

4. What quantum phases and what possible correlations at the electron/phonon levels can be obtained and how they contribute to/benefit the applications in Figs. 3 and 4 (as well as in supplementary figures)?

Response: We wish to clarify that the unique properties of electrons and phonons, and their interplay, equip the graphene superlattice to function in various capacities, including EM modulation,

luminescence enhancement, infrared stealth, and thermoelectric applications. For a more comprehensive exploration of these connections, we refer the reviewer to our detailed response to Comment 3 above.

5. How do the superlattice structures impact the characterized properties and related applications shown in this paper?

Response: We wish to clarify that the graphene superlattice, based on bilayer graphene, possesses a distinct structural framework:

- (i) Both the upper and lower layers feature an organized arrangement of nanometer-sized square nanopores.
- (ii) Particularly noteworthy, the alignment of each nanopore in the upper layer corresponds purposefully with those in the bottom layer.

This specific configuration leads to a unique characteristic: the presence of periodic electronic domain walls within a single atomic layer. This formation significantly impacts the behavior and interaction of electrons and phonons, enabling a wide range of applications, including EM wave modulation, fluorescence, and thermoelectric functions. For a more comprehensive understanding, we refer the reviewer to our detailed response to Comment 3 above.

Reviewer #2:

The authors have used decomposition and annealing method to generate nanopores in graphene layers and doped with guest atoms to endow graphene with multiple functions. However, before further consideration, the following concerns are suggested to be resolved.

Response: We thank the reviewer for their careful reading, and we have addressed the reviewer's comments below.

1) Bilayer graphene cannot be called as “superlattice”, is there any more proof? As superlattice materials are multilayer films grown alternately by two different components in thin layers ranging from a few nanometers to tens of nanometers and maintaining strict periodicity. We need to see the periodicity in the sectional dimension. Cs-TEM is needed to get precise information.

Response: We wish to clarify that, according to current research and definitions, 2D material superlattices can generally be categorized into two types:

- (i) Stacked 2D superlattices: This classification involves the periodic stacking of distinct 2D atomic layers, resulting in a thin film. The architectural confirmation of such superlattices typically benefits from cross-sectional TEM imaging (as mentioned in *Nature*, 2020, 579:368; *Nature*, 2022, 609:46).
- (ii) Uniform 2D superlattices: This type encompasses the assembly of identical 2D atomic layers, as seen in examples like ‘magic-angle graphene,’ which involves specific-angle twisting of two layers (as noted in *Nature*, 2018, 556:43; *Phys. Rev. B* 2018, 9:235453).

The superlattice we introduce aligns more closely with the latter category. Rather than relying on the stacking of various 2D atomic layers, our design is based on bilayer graphene with a specific architecture. It features high-density nanometer-sized square pores in the upper layer and a staggered arrangement in the lower layer, while retaining the intrinsic hexagonal atomic ring structure of graphene. While we acknowledge the value of a cross-sectional view, we believe that a top-down TEM perspective better captures the unique nanopore periodicity and finer details of our design.

2) From the measurement part, thermal conductivity seems to be measured in the out-plane direction. It is not in line with the electrical properties. The measured thermal conductivity in the out-plane direction after hot-pression will be much lowered, causing too high ZT, but not true. Please check.

Response: We thank the reviewer for their insightful comment. First, we wish to clarify our original methodology of determination of thermal conductivity, electrical conductivity and Seebeck coefficient:

- (i) Thermal conductivity: We measured the thermal conductivity by creating a 20 mm-diameter, 5.0 mm-thick disk from graphene superlattice powder via hot-pressing. The heat flow was directed across the disk's cross-section. Given the random orientation of the graphene nanosheets within the disk, the derived thermal conductivity does not represent either in-plane or out-of-plane direction.
- (ii) Seebeck coefficient and electrical conductivity: We produced a graphene superlattice film around 0.20 ± 0.02 mm thick from the bilayer graphene powder using vacuum filtration. This technique utilizes a vacuum to hasten the flow of the solution through filter paper. The pressure gradient ensures the swift passage of the solution, causing the graphene to accumulate on the paper, resulting

in a defined thickness, as evident in the cross-sectional image of the filtered film (see Fig. L1). The film orientation allowed measurements of values corresponding to its in-plane direction. However, deriving the ZT using our original methodology created inconsistencies due to the varying orientations.

Fig. L1. Schematic and SEM image showing the cross-section of the filtered bilayer graphene superlattice film.

To address the reviewer's comment, we have revised the assessment of the graphene superlattice's properties, particularly its in-plane and out-of-plane directions, employing novel methodologies.

(i) In-plane direction:

For a comprehensive evaluation of the in-plane thermoelectric characteristics, we employed the vacuum filtration technique to fabricate films with a specific orientation. Strips measuring 6 mm × 4 mm were crafted from these films. Approximately 15 of these strips, each 6 mm in length and 4 mm in width, were assembled to form a 3 mm-thick film. Electrodes were attached at both ends to facilitate measurements of in-plane electrical conductivity and the Seebeck coefficient along the film's length. To evaluate in-plane thermal conductivity, the filtered film was precisely sectioned into strips measuring 15 mm × 5 mm. About fifty of these strips were stacked to create a film approximately 10 mm thick, and in-plane thermal conductivity was measured with heat flow directed along the film's length.

(ii) Out-of-plane direction:

For measurements of out-of-plane electrical conductivity and the Seebeck coefficient, we stacked around a dozen filtered films to form a layered film with dimensions of 6 mm in length, 4 mm in width, and 3 mm in thickness, tailored to our equipment's testing parameters. The evaluation focused on the out-of-plane Seebeck coefficient and electrical conductivity orthogonal to the stacked layers. In assessing thermal conductivity, we utilized this film, comprising several ~0.2 mm thick layers and achieving an overall thickness of 5 mm. The out-of-plane thermal evaluation process ensured heat flow directed orthogonally through the plane of the film.

After the directional measurements of the in-plane and out-of-plane thermal conductivity, Seebeck coefficient, and electrical conductivity, the results are summarized below:

(i) In-plane performance:

The in-plane thermal conductivity of the superlattice ranges between 6.4–4.4 W/mK, representing

only 3.6–2.6% of the values observed in pristine graphene.

The Seebeck coefficient exhibits a notable increase of 710–560% compared to pristine graphene, while the electrical conductivity decreases by only 23.1% to 15.8%.

Notably, the in-plane ZT value for the superlattice reaches a significant peak of 0.33 at 500 K, clearly surpassing that of its unaltered counterpart.

(ii) Out-plane performance:

Thermal conductivity measures between 0.96 to 0.24 W/mK, with electrical conductivity recorded at 5.9–3.0 S/cm and the Seebeck coefficient varying between –7.5 to –12.0 $\mu\text{V/K}$.

The out-of-plane ZT value is observed to be 3–4 orders of magnitude lower than the in-plane value. This significant discrepancy underscores the dominant influence of in-plane properties on the overall thermoelectric performance, aligning with recent findings (*Adv. Funct. Mater.* 2020, 30:2003092; *Phys. Rev. Lett.* 2020, 125:226802).

To address the reviewer’s comment, we have added the following text to the revised manuscript:

“Characterization of in-plane thermal conductivity, electric conductivity and Seebeck coefficient of graphene superlattice

To obtain the in-plane Seebeck coefficient and electrical conductivity, a precise amount of bilayer graphene powder was dispersed in an ethanol solution. A film approximately 0.20 ± 0.02 mm thick was created using vacuum filtration. This filtered film was shaped into strips measuring 6 mm \times 4 mm in length and width. Around 15 of these strips were oriented and stacked to form a film with 3 mm thick. The Seebeck coefficient and electrical conductivity were evaluated along the length of the film using the ULVAC-RIKO ZEM-3 apparatus in a helium atmosphere at a heating rate of 5 $^{\circ}\text{C}/\text{min}$.

To assess the in-plane thermal conductivity, the filtered film was sectioned into strips measuring 15 mm in length and 5 mm in width. Approximately fifty of these strips were stacked to create a film approximately 10 mm thick. Heat conduction was directed along the length of the stacked film during the evaluation using the Netzsch HFM446 heat flow thermal instrument, ensuring precise measurement of in-plane thermal conductivity. Based on the in-plane Seebeck coefficient, electrical conductivity, and thermal conductivity, ZT were calculated as³⁶:

$$ZT = \sigma\alpha^2T/\kappa_T \quad (3)$$

where σ represents electrical conductivity, α denotes the Seebeck coefficient, κ_T corresponds to thermal conductivity, and T is the temperature.”

In addition, we have revised Fig. 4c-4e in the revised manuscript and added Supplementary Figs. 34 and 35 to the revised Supplementary Materials:

“

Fig. 4. Expanded applications of graphene superlattice. **a**, Tunable EM transmission of a graphene superlattice-based smart window via applied voltage. Photoluminescence intensity comparison between pristine graphene and graphene superlattice. **c**, **d**, Temperature-dependent in-plane thermal conductivity (κ_T) and in-plane Seebeck coefficient of pristine graphene and graphene superlattice. Error bars indicate standard deviations from three independent measurements. **e**, Comparison of in-plane figure merit values (ZT) of graphene superlattice with state-of-the-art thermoelectric materials. The graphene superlattice was Te-doped. The graphene superlattice achieved a maximum ZT of 0.33, even at a low temperature of 500 K, surpassing pristine graphene by four orders of magnitude and demonstrating comparability to leading inorganic thermoelectric materials.

Supplementary Fig. 34 – Electrical and thermal conductivity of pristine graphene and graphene superlattice. a, In-plane electrical conductivity. **b,** In-plane Seebeck coefficient. **c,** In-plane thermal conductivity. **d,** Out-of-plane electrical conductivity. **e,** Out-of-plane Seebeck coefficient. **f,** Out-of-plane thermal conductivity.

Supplementary Fig. 35 – Thermoelectric properties of pristine graphene and graphene superlattice. a, b, Figure of merit (ZT) values for (a) in-plane and (b) out-of-plane directions in graphene superlattice and pristine graphene.”

We have also revised the following text to the revised manuscript:

“Third, in the graphene superlattice, the presence of substantial coherent interference and phonon elastic scattering significantly diminishes in-plane lattice thermal conductivity, thereby leading to a decrease in the overall thermal conductivity, as evidenced in Fig. 4c, Supplementary Fig. 32 and Note 9. This thermal conductivity, ranging between 6.4–4.4 W/mK, constitutes only 3.6–2.6% of the values observed in pristine graphene. This exceptionally low thermal conductivity not only provides outstanding thermal insulation but also yields low infrared emissivity (~ 0.3 at the near-infrared band), making the graphene superlattice an ideal candidate for infrared stealth coatings (Supplementary Fig. 33). Furthermore, the combined effects of electron trapping and phonon drag significantly enhance the in-plane Seebeck coefficient of the graphene superlattice by 710–560% compared to pristine graphene in the temperature range of 300–500 K (Fig. 4d). Meanwhile, the electrical conductivity decreased by only 23.1–15.8% (Supplementary Fig. 34).

Current leading thermoelectric materials typically exhibit Seebeck coefficients within the 200–300 $\mu\text{V/K}$ range, only around two to four times the Seebeck coefficient of our developed graphene superlattice. However, they possess lower thermal conductivities, typically around 1.0 W/mK, significantly less than our graphene superlattice. It is important to note that typical graphene materials differ significantly from leading thermoelectric materials in their Seebeck coefficient and thermal conductivity - the former often being two orders of magnitude lower and the latter three to four orders of magnitude higher. Yet, our graphene superlattice leads in electrical conductivity, surpassing traditional graphene by almost an order of magnitude. This substantial difference accentuates the

exceptional thermoelectric performance of our graphene superlattice, resulting in remarkable in-plane figure-of-merit (ZT) values up to 0.33, even at a relatively modest temperature of 500 K (the effects on the out-of-plane parameters are detailed in Supplementary Figs. 34 and 35 and Note 10). This value is four orders of magnitude greater than pristine graphene, positioning our graphene superlattice among the most effective thermoelectric materials, as evidenced in Fig. 4e.”

3) As graphene is anisotropic, to make the conclusion convincing, both the in-plane and out-plane thermoelectric properties should be measured. It is supposed to be quite different. The effects of nanopores and Te-doping on the thermoelectric properties in different directions should be discussed in detail.

Response: To address the reviewer’s comment, we have performed a comprehensive analysis on our graphene superlattice samples in distinct directions. We refer the reviewer to our detailed response to Comment 2 above regarding the methodology.

We precisely determined the in-plane and out-of-plane ZT for the graphene. Figs. L2 and L3 reveal a significant disparity between the in-plane and out-of-plane ZT values in comparison to pristine graphene. Our observations showed that while the out-of-plane ZT closely approximated those of pristine graphene, the values were notably lower by 3 to 4 orders of magnitude in comparison to the strikingly higher in-plane ZT values. This discrepancy underlines the dominant influence of in-plane properties on the thermoelectric performance of the graphene superlattice. The structural alterations in the superlattice, notably the introduction of nanopores and Te element doping, significantly enhanced the in-plane thermoelectric performance while exhibiting minimal influence on out-of-plane properties.

Our focus on in-plane thermoelectric enhancement and relative limitations in the out-of-plane direction primarily centered on key structural modifications. Unlike traditional methods that modify interlayer interactions through small molecules, ions, or layer orientation, our approach strategically created nanopores and applied selective edge doping. These modifications notably transformed the in-plane electron and phonon structures, leading to the formation of electron traps, coherent interference, elastic phonon scattering, and robust electron–phonon coupling. Collectively, these changes contributed to a substantial increase in the in-plane ZT value. Conversely, in out-of-plane properties, the transport of electrons and phonons within the layers was primarily influenced by van der Waals interactions. Structural adjustments, including nanopores and Te-doping, had a limited impact on these interlayer forces, resulting in only modest improvement in out-of-plane thermoelectric properties compared to pristine graphene.

Fig. L2. In-plane and out-of-plane ZT of graphene superlattice film.

Fig. L3. ZT values for (a) out-of-plane and (b) in-plane directions in both graphene superlattice and pristine graphene.

To address the reviewer’s comment, we have added Supplementary Note 10 to the revised Supplementary Materials:

“Supplementary Note 10: Influence of graphene superlattice on in-plane and out-of-plane thermoelectric properties

To assess the influence of the graphene superlattice on both in-plane and out-of-plane directions, we conducted measurements of various thermoelectric properties, including electrical conductivity, Seebeck coefficient, and thermal conductivity (Supplementary Figs. 34 and 35).

i) In-plane direction:

For a comprehensive evaluation of the in-plane thermoelectric characteristics, we employed the vacuum filtration technique to fabricate films with a specific orientation. Strips measuring 6 mm × 4 mm were crafted from these films. Approximately 15 of these strips, each 6 mm in length and 4 mm in width, were assembled to form a 3 mm-thick film. Electrodes were attached at both ends to facilitate measurements of in-plane electrical conductivity and the Seebeck coefficient along the film’s length. To evaluate in-plane thermal conductivity, the filtered film was precisely sectioned into strips measuring 15 mm × 5 mm. About fifty of these strips were stacked to create a film approximately 10 mm thick, and in-plane thermal conductivity was measured with heat flow directed along the film’s length.

ii) Out-of-plane direction:

For measurements of out-of-plane electrical conductivity and the Seebeck coefficient, we stacked around a dozen filtered films to form a layered film with dimensions of 6 mm in length, 4 mm in width, and 3 mm in thickness, tailored to our equipment's testing parameters. The evaluation focused on the out-of-plane Seebeck coefficient and electrical conductivity orthogonal to the stacked layers. In assessing thermal conductivity, we utilized this film, comprising several ~0.2 mm thick layers and achieving an overall thickness of 5 mm. The out-of-plane thermal evaluation process ensured heat flow directed orthogonally through the plane of the film.

The results are summarized below:

i) In-plane performance:

In-plane thermal conductivity for the superlattice ranged between 6.4–4.4 W/mK, representing only 3.6–2.6% of pristine graphene's values. The Seebeck coefficient exhibited a remarkable increase of 710–560% compared to pristine graphene, while the electrical conductivity decreased by only 23.1% to 15.8%. The in-plane ZT value for the superlattice peaked at 0.33 at 500 K, significantly surpassing that of pristine graphene.

ii) Out-of-plane performance:

Out-of-plane thermal conductivity measured between 0.96 W/mK to 0.24 W/mK, with electrical conductivity recorded at 5.9–3.0 S/cm and the Seebeck coefficient varying between $-7.5 \mu\text{V/K}$ to $-12.0 \mu\text{V/K}$. These measurements closely aligned with those of pristine graphene, resulting in out-of-plane ZT values that were 3 to 4 orders of magnitude lower compared to the in-plane values.

Our investigation into the remarkable in-plane thermoelectric enhancement and relatively limited effects in the out-of-plane direction focused on pivotal structural modifications. These include the strategic creation of nanopores and selective edge doping within the plane, differing from conventional methods that alter interlayer interactions. Our synthetic method significantly modified in-plane electron and phonon structures, creating phenomena such as electron traps, coherent interference, elastic phonon scattering, and robust electron–phonon coupling. These collective alterations notably elevated the in-plane ZT value. While exploring interlayer dynamics, adjustments such as nanopore introduction or element doping had limited impact on fundamental interlayer forces, resulting in only a slight improvement in out-of-plane thermoelectric properties when compared to pristine graphene.”

4) Wondering how the electric conductivity and Seebeck coefficient of the ultra-thin graphene was measured on ZEM-3. And what is the direction? In-plane or out-plane?

Response: We wish to clarify that electrical conductivity and Seebeck coefficient measurements for the graphene superlattice were conducted along both in-plane and out-of-plane orientations using a ZEM-3 apparatus. We refer the reviewer to our detailed response to Comment 2 above regarding the methodology. Our methodology distinctly outlines that the material tested is not the usual multilayer graphene, which typically consists of stacked single atomic layers forming multilayer graphene or graphite. Rather, it constitutes a film derived from bilayer graphene via a filtration process.

5) For Fig 3b, is there an upper limit of the EM ab.%, as it was increased with thickness. What is the

density of the superlattice films? How about the curve of EM ab.% vs. density? "graphene" with thickness of 50 μm cannot be called graphene, it is graphite.

Response: We wish to clarify that in our investigation, we assessed EM absorption using a vector network analyzer with various thicknesses of graphene superlattice films fabricated through vacuum filtration. To address the reviewer's comment, we have extended our exploration to thicknesses ranging from 50 μm to 100 μm . Our findings revealed distinct stages in the absorption ratio as thickness increased, progressing through rapid increase, gradual rise, and finally reaching a saturation point at a maximum absorption value of about 62.1%. Regarding the correlation between EM absorption and density, our analysis involved graphene superlattice films of different thicknesses produced via filtration, using the graphene superlattice powder. Consequently, changes in thickness did not notably affect the film's density, maintaining approximately 8.6 g/cm^3 . This trend indicates no clear link between EM absorption and density. Our film production method, using a layer-by-layer filtration of bilayer graphene powder, yielded films predominantly composed of pure graphene, devoid of additional matrices. Consequently, an increase in film thickness led to a corresponding increase in density. Lastly, we wish to clarify that the mentioned 50 μm thickness does not refer to a single graphene superlattice nanosheet. Instead, it represents the cumulative thickness of a film composed of stacked bilayer graphene superlattice nanosheets formed through a filtration process.

To address the reviewer's comment, we have added the following text and Fig. 3b to the revised manuscript:

“This dispersive behavior leads to an absorption efficiency of graphene superlattice in the 2–5 GHz region, within the thickness range of 5–100 μm , that surpasses that of pristine graphene by two orders of magnitude (Fig. 3b). Even at a thickness of only 50 μm , the absorption can exceed 60%. We note here that the density remained consistent across varying thicknesses, maintaining a value of 8.6 g/cm^3 , underscoring the material's inherent lightweight nature.

Fig. 3. Permittivity and EM wave modulation of graphene superlattice. **a**, Frequency-dependent permittivity of graphene superlattice. **b**, Thickness-dependent EM absorption efficiency of pristine bilayer graphene and graphene superlattice. **c**, Comparison of EM shielding effectiveness (SE_T) and EM absorption efficiency ($A\%$) averaged by EM absorbing material thickness of graphene superlattice with carbon nanotubes (CNTs) and other 2D materials. **d**, Experimental setup of graphene superlattice-based EM–electricity converter. **e**, **f**, (e) Open circuit voltage (U_{oc}) and (f) EM–electricity conversion of pristine graphene and graphene superlattice-based EM–electricity converter as a function of EM emission power. The EM-to-electricity conversion efficiency is defined as the ratio of power generated by a graphene device, which converts input EM energy into direct current, to the input EM power. Error bars represent standard deviations from three independent measurements. **g**, Photographs of EM–electricity conversion by graphene superlattice-based EM–electricity converter from a 5G cell phone, a Wi-Fi router, a telecommunication tower, and a microwave (Supplementary Movies 1–4). The output DC electricity can be used to power electronic devices such as a digital hydro-thermometer and a collection of LEDs arranged in an ‘OSU’ pattern. The graphene superlattice was Te-doped.”

6) The thermal conductivity is incredibly low, please give information of the electrical conductivity and lattice thermal conductivity to make it convincing.

Response: To address the reviewer’s comment, we have performed additional characterization and analysis of thermal conductivity of graphene superlattice. We have added Supplementary Note 9 to the revised Supplementary Materials:

“Supplementary Note 9: Calculation of electron and lattice thermal conductivity in graphene superlattice

To comprehensively analyze the thermal conductivity (κ_T) mechanism in the device, a calculation based on the Widemann–Franz relationship allows the determination of lattice (κ_L) and electron thermal conductivity (κ_e)⁷⁹:

$$\kappa_T = \kappa_e + \kappa_L \quad (S7)$$

$$\kappa_e = \sigma \times L \times T \quad (S8)$$

where L represents the Lorenz number (in units of $10^{-8} \text{ W}\Omega\text{K}^{-2}$), The Lorenz number L has a proportional relationship with the Seebeck coefficient α , outlined as⁸⁰:

$$L = 1.5 + ex \left(-\frac{|\alpha|}{116} \right) \quad (S9)$$

Utilizing the above equations, the in-plane values for κ_e and κ_L can be determined at increasing temperatures. Upon analysis, it becomes apparent that in-plane κ_L predominantly governs the overall thermal conductivity (Supplementary Fig. 32). In comparison to pristine graphene, a significant reduction in in-plane κ_L by approximately 98% stands out as the primary factor contributing to the observed diminished thermal conductivity. This substantial decrease in in-plane κ_L is intricately linked to alterations in the phonon structure. Pristine graphene mainly exhibits atom-propagated phonons oscillating at a frequency of 50 THz, as depicted in Fig. 2c. This behavior stems from the harmonic oscillation inherent in the atomic lattice. In contrast, the periodic overlap of carbon atoms in bilayer graphene superlattices suppresses these harmonic vibrations, resulting in the emergence of distinct phonon modes in the overlapped and exposed regions. The transmission of these phonon clusters exhibits considerable coherent interference and elastic scattering. Consequently, phonon cluster energy weakens and a series of continuous standing waves arise in the low-frequency range of 10–30 THz, while the original high-frequency phonons diminish. These alterations are directly attributed to the observed reduction in in-plane thermal conductivity.”

In addition, we have added Supplementary Fig. 32 to the revised Supplementary Materials:

Supplementary Fig. 32 – Temperature-dependent in-plane thermal conductivity of pristine graphene and graphene superlattice film. a, lattice thermal conductivity. b, electron thermal conductivity.”

Reviewer #3:

Currently, it is a great challenge for fabrication of 2D superlattice structures composed of chemically modified 2D material sublattices, especially the exploration of 2D superlattice structures in nanoporous graphene. In this work, the authors reported a chemical method to synthesize a family of 2D superlattice structures to tailor the electronic and phonon structure of superlattice, through introducing additional degrees of freedom. This is an interesting and novel work that deserves publication in Nature Communications. However, there are still a couple of issues needed to note and it may be published after major revision. The comments are as follows:

Response: We thank the reviewer for their careful reading of our manuscript and for noting that our work is interesting and novel and it deserves publication in Nature Communications.

1. What is the advantage of graphene super-lattice for the ZT factor? Especially, many materials have the ZT factors larger than 1.6 such as Lidong Zhao et al. published papers in Science. The authors did not add these data in the figure.

Response: To address the reviewer's comment, we have included a comparative ZT value analysis in the revised manuscript, showcasing impressive figures from various high-performance inorganic thermoelectric materials. For instance, $\text{SnS}_{0.91}\text{Se}_{0.09}$ recorded a peak ZT of 1.61 at 873 K, Pd-based materials reached 1.61 at 773 K, Ga-doped PbS achieved 0.84 at 923 K, $\text{Eu}_2\text{Zn}_{0.98}\text{Sb}_2$ exhibited a ZT of 1.01 at 823 K, and AgSbTe_2 delivered an impressive ZT value of 1.51 at 573 K. Comparatively, our engineered graphene superlattice has achieved a noteworthy ZT value of 0.33 at a moderate temperature of 500 K. This performance not only significantly outperforms pristine graphene but also represents an enhancement of 3–4 orders of magnitude. Furthermore, a direct comparison with ZT values of prominent inorganic thermoelectric materials highlights the narrowing disparity, particularly at low temperatures (between 300 K and 500 K). This underscores the substantial potential of our graphene superlattice in the field of thermoelectric applications.

We have added the following text and revised Fig. 4 in the revised manuscript:

“This substantial difference accentuates the exceptional thermoelectric performance of our graphene superlattice, resulting in remarkable in-plane figure-of-merit (ZT) values up to 0.33, even at a relatively modest temperature of 500 K (the effects on the out-of-plane parameters are detailed in Supplementary Figs. 34 and 35 and Note 10). This value is four orders of magnitude greater than pristine graphene, positioning our graphene superlattice among the most effective thermoelectric materials, as evidenced in Fig. 4e.

Fig. 4. Expanded applications of graphene superlattice. **a**, Tunable EM transmission of a graphene superlattice-based smart window via applied voltage. Photoluminescence intensity comparison between pristine graphene and graphene superlattice. **c**, **d**, Temperature-dependent in-plane thermal conductivity (κ_T) and in-plane Seebeck coefficient of pristine graphene and graphene superlattice. Error bars indicate standard deviations from three independent measurements. **e**, Comparison of in-plane figure merit values (ZT) of graphene superlattice with state-of-the-art thermoelectric materials. The graphene superlattice was Te-doped. The graphene superlattice achieved a maximum ZT of 0.33, even at a low temperature of 500 K, surpassing pristine graphene by four orders of magnitude and demonstrating comparability to leading inorganic thermoelectric materials.”

In addition, we have added the following references to the revised Supplementary Materials:

“122. He, W.K. *et al.* High thermoelectric performance in low-cost SnS_{0.91}Se_{0.09} crystals. *Science* **365**, 1418 (2019).

123. Xiao, Y. *et al.* Ultrahigh carrier mobility contributes to remarkably enhanced thermoelectric performance in *n*-type PbSe. *Energy Environ. Sci.* **15**, 346 (2022).

124. Chen, Z. X. *et al.* GaSb doping facilitates conduction band convergence and improves thermoelectric performance in *n*-type PbS. *Energy Environ. Sci.* **16**, 1676 (2023).

125. Chen, C. *et al.* Zintl-phase Eu₂ZnSb₂: a promising thermoelectric materials with ultralow thermal conductivity. *Proc. Natl. Acad. Sci.* **116**, 2831 (2019).

126. Subhajib, R. *et al.* Enhanced atomic ordering leads to high thermoelectric performance in AgSbTe₂. *Science* **371**, 722 (2021).”

2. I cannot see the better performance of thermoelectricity for the reported graphene superlattice including the various thermoelectric factors such as Seebeck coefficient and so on. They need to compare most of the best materials currently published online.

Response: To address the reviewer’s comment, we aim to elucidate the remarkable performance of our

graphene superlattice concerning leading inorganic thermoelectric materials, as detailed in our response to Comment 1. The graphene superlattice achieves an impressive ZT value of 0.33 at 500 K, outperforming pristine graphene by an exceptional 3–4 orders of magnitude. In comparison to prevalent inorganic thermoelectric materials, particularly within the 300–500 K temperature range, the performance gap substantially diminishes. These results highlight the promising potential of our graphene superlattice in thermoelectric applications, reducing the once substantial disparity to a modest difference compared to contemporary inorganic thermoelectric materials.

For a deeper understanding of this enhanced performance, we conducted a comprehensive analysis, comparing key thermoelectric parameters such as electrical conductivity, thermal conductivity, and the Seebeck coefficient with high-performance inorganic materials. At the peak ZT , our electrical conductivity reaches approximately 5,000 S/cm, the Seebeck coefficient stands at about 70 $\mu\text{V}/\text{K}$, and the thermal conductivity is close to 4.4 W/mK. As depicted in Fig. 4e of the revised manuscript, leading thermoelectric materials generally exhibit a Seebeck coefficient in the range of 200–300 $\mu\text{V}/\text{K}$, nearly 3–4 times more pronounced than our graphene superlattice. Their thermal conductivity, approximately 1.0 W/mK, is also lower than that of graphene superlattice. However, it is noteworthy that their electrical conductivity lags behind our graphene superlattice by nearly an order of magnitude.

Current graphene materials deviate significantly from mainstream thermoelectric materials in terms of Seebeck coefficient and thermal conductivity, being two orders of magnitude lower and three to four orders of magnitude higher, respectively. Nevertheless, our graphene superlattice material has remarkably bridged this gap, narrowing these differences to a mere multiple, all while maintaining outstanding electrical conductivity, leading to a ZT value comparable to leading inorganic thermoelectric materials at low and intermediate temperatures. These findings lead us to believe that the integration of graphene superlattices holds promising avenues for further advancements in thermoelectric research, leveraging the distinct attributes of these organized materials for more efficient energy applications.

To address the reviewer's comment, we have added the following text to the revised manuscript:

“Current leading thermoelectric materials typically exhibit Seebeck coefficients within the 200–300 $\mu\text{V}/\text{K}$ range, only around two to four times the Seebeck coefficient of our developed graphene superlattice. However, they possess lower thermal conductivities, typically around 1.0 W/mK, significantly less than our graphene superlattice. It is important to note that typical graphene materials differ significantly from leading thermoelectric materials in their Seebeck coefficient and thermal conductivity - the former often being two orders of magnitude lower and the latter three to four orders of magnitude higher. Yet, our graphene superlattice leads in electrical conductivity, surpassing traditional graphene by almost an order of magnitude. This substantial difference accentuates the exceptional thermoelectric performance of our graphene superlattice, resulting in remarkable in-plane figure-of-merit (ZT) values up to 0.33, even at a relatively modest temperature of 500 K (the effects on the out-of-plane parameters are detailed in Supplementary Figs. 34 and 35 and Note 10). This value is four orders of magnitude greater than pristine graphene, positioning our graphene superlattice among

the most effective thermoelectric materials, as evidenced in Fig. 4e.”

3. Please check all the full names of the abbreviations given when they first appeared. Besides, all the full names should be uniform. There are numerous lines in most figures, when jointing the images. Turn off borders on images, and remove the redundant lines.

Response: We thank the reviewer for their careful reading. We have reviewed the manuscript to ensure all abbreviations are adequately introduced with their full names upon their initial mention and have maintained their consistent use throughout the document. Concerning the figures, we have taken the necessary steps to remove any extraneous lines and disabled borders to enhance the visual clarity of the images.

4. In the Introduction part, when reviewing the background of 2D superlattices, more classic references should be cited, instead of citing papers published in the last two years.

Response: To address the reviewer’s comment, we have added the following references on 2D superlattices to the revised manuscript:

“[2] Cao, Y. *et al.* Unconventional superconductivity in magic-angle graphene superlattices. *Nature* **556**, 43 (2018).

[5] Chen, P. *et al.* Chemical synthesis of two-dimensional atomic crystals, heterostructures and superlattices. *Chem. Soc. Rev.* **467**, 3129 (2018).

[7] Zhang, Z. W. *et al.* Robust epitaxial growth of two-dimensional heterostructures, multi-heterostructures, and superlattices. *Science* **357**, 788 (2017).

[9] Geuchies, J. J. *et al.* In-situ study of the formation mechanism of two-dimensional superlattices from PbSe nanocrystals. *Nat. Mater.* **15**, 1248 (2016).”

REVIEWER COMMENTS

Reviewer #1 (Remarks to the Author):

1. In response to Reviewer #2, Question 1, the authors argue that: "It features high-density nanometer-sized square pores in the upper layer and a staggered arrangement in the lower layer." However, no direct evidence shows any structural or physical difference between the upper layer and lower layer. More experimental evidence is needed to prove the heterogeneity and differences between two layers, including structural analysis of the two layers and high-resolution TEM imaging.
2. In response to Reviewer #1, Question 1, the authors argue that: "Unfortunately, we observed irregular ellipsoidal and polyhedral formations in both CoO and NiO, deviating significantly from our intended uniform structure." This indicates that the size and dispersity of the template would induce different size, distortion or morphology of the nanopores. How did these factors affect the synthesis of bilayer graphene superlattices?
3. Why is Te and the chosen elements employed for doping porous graphene superlattices? What are the fundamental and technical benefits? Could other materials be used for the same purpose?
4. Why would using Fe₃O₄ nanoparticles as a template induce misaligned nanopores in bilayer graphene while using NaCl would induce overlapped nanopores? The authors should discuss the formation mechanism of misaligned nanopores in more detail.
5. The authors should discuss the purpose of introducing dopants into the graphene superlattice more. Also, are the "graphene superlattice" in fig. 3 and fig. 4 doped graphene superlattice? If so, the authors should also include the comparison of undoped graphene superlattice.

Reviewer #2 (Remarks to the Author):

The authors have indeed made an effort to address certain issues in their work. However, some questions still remain unanswered, and further clarification is needed on specific points:

1. The authors claim that the thermal conductivity derived from graphene disks does not represent either in-plane or out-of-plane direction due to the random orientation of graphene nanosheets. This statement seems contradictory, especially considering the potential preferential-orientation caused by hot-pressing. Could the authors elaborate further to reconcile this apparent contradiction?
2. Figure 34b and e show a significant difference in the in-plane and out-plane Seebeck coefficients. Generally, Seebeck is primarily carrier concentration dependent, the observed variation is unusual. Could the authors explain the reason behind this significant difference?
3. Figure 34e and f present a conflict with the common rule. Generally, the out-plane thermal conductivity of graphene is much lower than in-plane one. However, the Figure 34e and f present a opposite phenomenon. Could the authors provide an explanation for this discrepancy?
4. To ensure a convincing ZT value, it is crucial for the authors to verify and address the above-discussed points.

Reviewer #1:

1. In response to Reviewer #2, Question 1, the authors argue that: “It features high-density nanometer-sized square pores in the upper layer and a staggered arrangement in the lower layer.” However, no direct evidence shows any structural or physical difference between the upper layer and lower layer. More experimental evidence is needed to prove the heterogeneity and differences between two layers, including structural analysis of the two layers and high-resolution TEM imaging.

Response: We wish to clarify here that the heterogeneity present in our graphene superlattice materials systems primarily stems from nanometer-sized pores, rather than the variations in carbon atom arrangements within the graphene layers. In our previous revision, we elucidated that, by definition, 2D superlattices are typically classified into the following two categories:

- a) *Heterogeneous 2D superlattices:* This classification involves the periodic stacking of distinct 2D atomic layers, resulting in a thin film (as mentioned in *Nature*, 2020, 579:368; *Nature*, 2022, 609:46).
- b) *Homogeneous 2D superlattices:* This type encompasses the assembly of identical 2D atomic layers, as seen in examples like ‘magic-angle graphene,’ which involves specific-angle twisting of two layers (as noted in *Nature*, 2018, 556:43; *Phys. Rev. B*, 2018, 9:235453).

To categorize our graphene superlattice, we outlined the synthetic process as depicted in Fig. L1:

Fig. L1 - Schematic of the synthesis process for bilayer graphene superlattice with partially overlapped nanopores.

- a) Initially, through the application of liquid-phase in-situ growth methods, we grow densely packed Fe_3O_4 nanoparticle templates, each with a precise diameter of approximately 7.0 nm, uniformly distributed across both the upper and lower surfaces of the bilayer graphene.
- b) The subsequent annealing process initiates a carbothermal reduction reaction between the Fe_3O_4 nanoparticles and the graphene’s surface carbon atoms, converting the carbon atoms in immediate contact with the templates into CO_2 gas. This process leads to the formation of ordered nanopores, with their sizes and shapes templated by the Fe_3O_4 nanoparticles.
- c) Finally, we employ an acid etching technique to remove these templates. This process generates bilayer graphene with precisely distributed nanometer-scale square pores intricately patterned across its entirety. The asymmetrical distribution of Fe_3O_4 templates induces misalignment,

resulting in a characteristic periodic offset in pore structures between the upper and lower layers. The partially overlapped porous structure forms a unique arrangement of carbon atoms with a periodicity of 3.2 ± 0.7 nm, as evident in the inset of Fig. 1d of the main text.

Depending on the developed graphene superlattice, we wish to clarify that our bilayer graphene superlattice is featured with high-density, monodisperse nanometer-sized square pores on both the upper- and lower-layer surfaces. Notably, the nanopores in the upper layer are partially overlapped with those in the lower layer. This structural feature is distinct from the periodic displacement of carbon atoms observed in systems like twisted magic-angle graphene. Hence, we believe that our graphene superlattice falls into the heterogeneous 2D superlattice, which the “heterogeneity” refers to the periodic partially overlapped pore structures.

In order to characterize the heterogeneity, as suggested by the reviewer, we utilized advanced high-resolution TEM. Specifically, we employed low-voltage high-angle annular dark field scanning transmission electron microscopy (HAADF-STEM) to achieve detailed and precise characterizations. However, our bilayer graphene superlattice, which possesses partially overlapped nanopores, results in a surface that lacks perfect flatness. posed a substantial obstacle to achieving atomic-resolution imaging in TEM analyses. Theoretically, even if atomic-level resolution were achieved, the specific alignment of carbon atoms between the upper and lower layers in our structure prevents the observation of distinct phenomena, such as Moiré fringes, in our investigation. In response to limitations, we switched to selected area electron diffraction (SAED) to compare our bilayer graphene superlattice with a synthesized monolayer sublattice. The SAED image revealed an atypical ring formation, indicative of amorphous characteristics resulting from the elevated nanopore density in both single-layer and bilayer samples. Notably, aside from this amorphous trait, no discernible structural disparities were observed in our analysis, as depicted in Fig. L2.

Fig. L2 - SAED analysis of (a) graphene superlattice and (b) its corresponding sublattice which consists of an ordered array of square-shaped, nanometer-sized porous monolayer graphene. This sublattice was fabricated using the same methodology as the graphene superlattice.

2. In response to Reviewer #1, Question 1, the authors argue that: “Unfortunately, we observed irregular ellipsoidal and polyhedral formations in both CoO and NiO, deviating significantly from our intended uniform structure.” This indicates that the size and dispersity of the template would induce different

size, distortion or morphology of the nanopores. How did these factors affect the synthesis of bilayer graphene superlattices?

Response: We would like to clarify that in the previous version of our study, specifically detailed in Supplementary Fig. S3, we conducted a systematic examination of various template conditions and their subsequent effects on the synthesis process of the bilayer graphene superlattice:

- a) The synthesis of porous graphene through the in-situ growth of Fe_3O_4 templates on wrinkled reduced graphene oxide, as depicted in Supplementary Fig. 3a and 3b.
- b) Inadequate and excessive durations of carbon thermal reduction between the template and graphene resulted in distinct outcomes as shown Supplementary Fig. 3c and 3d.
- c) Graphene synthesis involving the overloading of templates resulted in template aggregation and poor dispersion, as shown in Supplementary Fig. 3e and 3f.
- d) The effects of excessive annealing temperatures are shown in Supplementary Fig. 3g and 3h.

Supplementary Fig. 3 – Morphology of reduced graphene oxide and porous graphene. a-h, Representative TEM images of (a) wrinkled pristine reduced graphene oxide, (b) porous reduced graphene oxide nanosheets obtained by annealing cubic Fe_3O_4 nanoparticle-coated reduced graphene oxide, (c, d) porous graphene annealed at 650 °C for (c) 15 minutes and (d) 1 hour, (e, f) porous graphene annealed at obtained by incorporating an excessive quantity of cubic Fe_3O_4 nanoparticles (the loading amount of Fe_3O_4 nanoparticles is approximately 1.2 times greater than the amount used during the production of graphene superlattice), (g, h) porous graphene obtained by annealed at 1,000 °C and (g) 800 °C for 30 minutes. The observations reveal that insufficient annealing time results in incomplete reduction between the cubic Fe_3O_4 nanoparticle template and carbon atoms, hindering the formation of monodisperse nanopores on the graphene surface. Prolonged reaction time leads to excessive reduction and polydisperse and distorted pores, as the cubic Fe_3O_4 nanoparticle template continues to react with carbon on the second layer. High annealing temperature or excessive loading amounts causes cubic Fe_3O_4 nanoparticle template recrystallization and agglomeration, resulting in polydisperse and distorted pores. Precise control of reaction time, temperature and loaded amount is crucial for achieving monodisperse, square nanopores.

A consistent conclusion from the above studies indicates shortcomings in achieving the precise creation of organized nanometer-sized nanopores on the graphene surface.

3. Why is Te and the chosen elements employed for doping porous graphene superlattices? What are the fundamental and technical benefits? Could other materials be used for the same purpose?

Response: We wish to clarify that past research has identified elements such as tellurium (Te) as frequently used dopants for graphene, which have been shown to effectively alter graphene's electronic structure (*Nano Energy*, 2016, 30:867; *Sci. Bull.* 2020, 65:1580), indirectly influencing its physical properties, including the lower-frequency permittivity that we have emphasized in our research. Besides Te, our bilayer graphene superlattice has also been doped with other commonly used elements such as boron (B; referenced in *Joule* 2018, 2, 1610), nitrogen (N; referenced in *Adv. Mater.* 2021, 33:2003521), sulfur (S; referenced in *J. Am. Chem. Soc.* 2017, 139:4506), phosphorus (P; referenced in *Energy Environ. Sci.* 2017, 10:116), and selenium (Se; referenced in *Angew. Chem. Int. Ed.* 2018, 130:4772). Our results demonstrate the ability to dope various types of dopants on the pore edges of our graphene superlattice, indicating the generalizability of our approach. We believe that our method can extend to other commonly employed elements for doping graphene and other carbon nanomaterials.

The strategic placement of these dopants at the pore edges within the graphene superlattice structure can create dipoles whose polarity can be precisely controlled. This particular configuration not only markedly boosts the electromagnetic wave within the low-frequency spectrum, but also demonstrates a modified effective absorption frequency in response to external electromagnetic field exposure. Leveraging this concept, we are able to attain maximum peak absorption of electromagnetic waves, surpassing 60% efficiency across various frequency bands through strategic replacement of the doping element. This effect is clearly depicted in the Supplementary Fig. S26 of our previous version of Supplementary Materials.

Supplementary Fig. 26 – Doped element-dependent EM absorption of graphene superlattice. The maximum absorption frequencies for different dopants in graphene superlattice were determined as follows: H-doped (3.2 GHz), B-doped (3.9 GHz), N-doped (4.3 GHz), S-doped (2.9 GHz), O-doped (3.5 GHz), P-doped (4.6 GHz), and Se-doped (2.8 GHz). Notably, all of these dopants exhibited EM absorption efficiencies exceeding 60%. This remarkable feature enables the graphene superlattice to selectively absorb EM waves at a specific wavelength while allowing the transmission of other frequency bands, thereby minimizing unnecessary signal loss. Error bars represent standard deviations

from three independent measurements.

To address the reviewer's comment, we have added the following text to the revised manuscript:

“We believe our method has the potential to extend to other commonly employed elements for doping graphene and other carbon nanomaterials, enabling precise tuning of their electronic and phonon structures.”

4. Why would using Fe_3O_4 nanoparticles as a template induce misaligned nanopores in bilayer graphene while using NaCl would induce overlapped nanopores? The authors should discuss the formation mechanism of misaligned nanopores in more detail.

Response: We wish to clarify that we achieved the misalignment of nanopores by depositing Fe_3O_4 templates on both surfaces of the bilayer graphene, followed by a subsequent carbothermal reaction and etching step. For further details, we direct the reviewer to our response to Comment 1. Below, we present the detailed steps involved in the creation of graphene with completely overlapped nanopores, as illustrated in Fig. L3:

Fig. L3 - Schematic of the synthesis process for graphene with completely overlapped nanopores.

- Initiated by submerging NaCl particles, ranging from 5 to 10 micrometers, in an oleic acid iron solution, this step ensures the adsorption of oleic acid iron onto the NaCl particles, resulting in a coated layer of oleic acid iron on the NaCl surface.
- In the thermal decomposition phase, the iron oleate on the NaCl surface undergoes a transformation into cubic Fe_3O_4 nanoparticles. Simultaneously, carbon sheets form, encapsulating these cubic nanoparticles and incorporating the Fe_3O_4 into carbon nanosheets.
- Subsequent to the thermal decomposition, the composite material is immersed in water to dissolve and remove the NaCl substrate. An acid wash follows, selectively extracting the Fe_3O_4 , resulting in the production of aligned, porous carbon nanosheets.
- The process concludes with a high-temperature reduction in a flow of hydrogen gas, effectively converting the carbon nanosheet into graphene with completely overlapped nanopores.

In this method, NaCl is employed as a crucial substrate for graphene growth but plays no direct role in pore formation. The genesis of these pores is closely associated with the decomposition of oleic acid iron, leading to the emergence of cubic Fe_3O_4 templates.

To address the reviewer's comment, we have added the following text to the revised manuscript.

“The partially overlapped pores are ascribed to the presence of misaligned monodisperse Fe₃O₄ nanoparticles at both the upper and lower layers.”

“We produced graphene with fully overlapped nanopores using a bottom-up approach involving the thermal decomposition of oleic acid iron coated sodium chloride (NaCl) particles. Initially, 0.1 g of NaCl particles, ranging from 5 to 10 μm, were submerged in 10 mL of oleic acid iron solution. This step ensures the adsorption of the oleic acid iron onto the NaCl particles, resulting a coated layer of oleic acid iron on the NaCl surface. During thermal decomposition at 500 °C with a nitrogen gas flow, the iron oleate on the NaCl surface undergoes a transformation into cubic Fe₃O₄ nanoparticles. Simultaneously, carbon sheets form, encapsulating these cubic nanoparticles and incorporating the Fe₃O₄ into carbon nanosheets. Subsequent to the thermal decomposition, the composite material is immersed in water to dissolve and remove the NaCl substrate. An acid wash follows, selectively extracting the Fe₃O₄, resulting in the production of aligned, porous carbon nanosheets. Lastly, graphene with completely overlapped nanopores was obtained by drying in a vacuum oven for 12 hours, followed by thermal reduction at 1,500°C for 2 hours in a hydrogen gas flow.”

5. The authors should discuss the purpose of introducing dopants into the graphene superlattice more. Also, are the “graphene superlattice” in fig. 3 and fig. 4 doped graphene superlattice? If so, the authors should also include the comparison of undoped graphene superlattice.

Response: In the research of graphene, elemental doping has emerged as a predominant approach for manipulating its electronic characteristics. This technique’s significance is underscored in seminal works such as those published in *Nano Energy* 2016, 30:867, *Sci. Bull* 2020, 65:15, *Adv. Mater.* 2021, 33:2003521, *J. Am. Chem. Soc.* 2017, 139:4506 and *Angew. Chem. Int. Ed.* 2018, 130:4772. For detailed explanations on the selection of Te and other elements as dopants, we direct the reviewer to our response to Comment 3.

Additionally, we wish to clarify that the graphene superlattice showcased in Figs. 3 and 4 are illustrations of the Te-doped graphene superlattices, as outlined in the captions of the figure presented in the previous version of the main text.

In response to the reviewer’s concerns about the comparison of undoped graphene superlattice with doped counterparts, we have conducted a comparative analysis of the permittivity performance between undoped and doped graphene superlattices. As shown in Fig. L4, while undoped porous graphene superlattice exhibits dielectric relaxation polarization peaks in the low-frequency range, it is evident that the polarization strength is significantly lower than that of Te-doped graphene superlattice. The reason for this weakened intensity lies in the fact that the edges of the pores in undoped porous graphene primarily consist of various forms of covalent bonds involving mixed carbon, oxygen, and hydrogen. This mixture diminishes the overall bond polarity. Due to this weakening effect, the maximum electromagnetic wave absorption rate of undoped porous graphene is only 21.2%, notably lower than that of Te-doped graphene superlattice.

Fig. L4 – Frequency-dependent permittivity of (a) Te-doped and (b) undoped graphene superlattice.

To address the reviewer’s comment, we have added the following text and Supplementary Fig. S24 to revised Supplementary Materials:

“We observed dielectric relaxation polarization peaks in the low-frequency range for undoped porous graphene. However, it is evident that the polarization strength is considerably lower than that of the doped graphene superlattice. This discrepancy is attributed to the existence of various weak polarity bonds, consisting of a mixture of carbon, oxygen, and hydrogen, at the pore edges.”

Supplementary Fig. 24 – Frequency-dependent permittivity of pristine graphene and graphene

superlattice. a, Pristine bilayer graphene. **b,** Undoped porous graphene superlattice. **c-i,** Doped porous graphene superlattice.”

Reviewer #2:

The authors have indeed made an effort to address certain issues in their work. However, some questions still remain unanswered, and further clarification is needed on specific points:

1. The authors claim that the thermal conductivity derived from graphene disks does not represent either in-plane or out-of-plane direction due to the random orientation of graphene nanosheets. This statement seems contradictory, especially considering the potential preferential-orientation caused by hot-pressing. Could the authors elaborate further to reconcile this apparent contradiction?

Response: To address the reviewer's concern regarding the thermal conductivity of graphene prepared by hot pressing, we carried out a comprehensive analysis using SEM. Our primary focus was on examining the cross-section of the graphene disks, as this aligns with the direction of heat flow. The SEM image revealed a non-uniform alignment of graphene nanosheets. Some nanosheets lay flat due to their larger surface areas, while others exhibited a random and disorganized arrangement, likely stemming from the pressing process. This heterogeneity in alignment suggests that, although there may be instances of preferential orientation during the pressing procedure, they are not the predominant feature in our samples. Consequently, the thermal conductivity values obtained from these graphene disks do not consistently represent either the in-plane or out-of-plane directions.

Fig. L5 – Representative SEM image of the cross-section of graphene superlattice disk prepared by hot-pressing.

2. Figure 34b and e show a significant difference in the in-plane and out-of-plane Seebeck coefficients. Generally, Seebeck is primarily carrier concentration dependent, the observed variation is unusual. Could the authors explain the reason behind this significant difference?

Response: We thank the reviewer for their careful reading. We mistakenly mix up data and images between the in-plane and out-of-plane direction interchanged data and images between the in-plane and out-of-plane directions. We agree with the reviewer that in-plane Seebeck coefficients and thermal conductivity of two-dimensional materials such as graphene are generally larger than those in the out-of-plane direction. To correct this error, we have revised Supplementary Fig. S34 in the revised Supplementary Materials.

Supplementary Fig. 34 – Electrical and thermal conductivity of pristine graphene and graphene superlattice. **a**, In-plane electrical conductivity. **b**, In-plane Seebeck coefficient. **c**, In-plane thermal conductivity. **d**, Out-of-plane electrical conductivity. **e**, Out-of-plane Seebeck coefficient. **f**, Out-of-plane thermal conductivity.

3. Figure 34e and f present a conflict with the common rule. Generally, the out-plane thermal conductivity of graphene is much lower than in-plane one. However, the Figure 34e and f present a opposite phenomenon. Could the authors provide an explanation for this discrepancy?

Response: We thank the reviewer for their careful reading. We have fixed this error and we refer the reviewer to our detailed response to Comment 2.

4. To ensure a convincing ZT value, it is crucial for the authors to verify and address the above-discussed points.

Response: To address the reviewer’s comment, we have revised Supplementary Fig. S35 in the revised Supplementary Materials:

Supplementary Fig. 35 – Thermoelectric properties of pristine graphene and graphene

superlattice. a, b, Figure of merit (ZT) values for (a) in-plane and (b) out-of-plane directions in graphene superlattice and pristine graphene.

REVIEWER COMMENTS

Reviewer #1 (Remarks to the Author):

The authors have addressed my previous comments.

Reviewer #2 (Remarks to the Author):

1. The authors say “We mistakenly mix up data and images between the in-plane and out-of-plane direction interchanged data and images between the in-plane and out-of-plane directions”. This shall be avoided for publication in NC.

2. Generally, Seebeck is carrier concentration dependent. How such significant difference was generated between the in-plane and out-plane direction (nearly 10:1 = out-plane:in-plane)? Please explain carefully. This is very important.

Reviewer 2

1. The authors say “We mistakenly mix up data and images between the in-plane and out-of-plane direction interchanged data and images between the in-plane and out-of-plane directions”. This shall be avoided for publication in NC.

Response: We thank the reviewer for their careful reading of our manuscript and we have corrected the mistake in the previous version of the manuscript.

2. Generally, Seebeck is carrier concentration dependent. How such significant difference was generated between the in-plane and out-plane direction (nearly 10:1 = out-plane:in-plane)? Please explain carefully. This is very important.

Response: We wish to clarify that we fixed an error in the Seebeck coefficient for in-plane and out-of-plane directions in Supplementary Fig. 34 of the previous version of Supplementary Information. It is important to note that the in-plane Seebeck coefficient exceeds that in the out-of-plane direction, by a factor of approximately 8.8 to 6.3 in the temperature range of 300–500 K.

We sought to understand the mechanism behind the difference between in-plane and out-of-plane Seebeck coefficients. Past research has indicated that the Seebeck coefficient of a 2D material is not only related to the carrier concentration but also largely influenced by carrier mobility, energy-dependent electrical conductivity, density of states, Fermi distribution function, and applied temperature. These factors can be collectively elucidated through the Mott equation (*Phys. Rev. Lett.* 2020, 125:117001):

$$S = \frac{\pi^2 k_B}{3 e} k_B T \left\{ \frac{d[\ln(\sigma(E))]}{dE} \right\}_{E=E_F} = \frac{\pi^2 k_B}{3 e} k_B T \left\{ \frac{1}{n} \frac{dn(E)}{dE} + \frac{1}{\mu} \frac{d\mu(E)}{dE} \right\}_{E=E_F} \quad (L1)$$

in which k_B is the Boltzmann constant, T is the temperature, E_F is the Fermi level, e is the electron charge, μ and n represent the carrier mobility and concentration, respectively. $\sigma(E)$, $n(E)$ and $\mu(E)$ denote the energy-dependent electrical conductivity, carrier concentration, and carrier mobility, respectively, which are intricately associated with the band structure of 2D materials. $n(E)$ is determined by:

$$n(E) = g(E)f(E) \quad (L2)$$

where $g(E)$ is the density of states and $f(E)$ is the Fermi distribution function. It is worth mentioning that the applicability of the Mott equation is specific to semiconductors and metals, and it is not suitable for electrical insulators (*Energy Environ. Sci.* 2012, 5:5510).

The Mott equation above implies that, despite lower carrier concentrations, the Seebeck coefficient in the out-of-plane direction of 2D materials can be lower than that in the in-plane direction. For instance, trilayer 2D material SnS₂ has an in-plane carrier concentration of 10^{18} cm^{-3} with an in-plane Seebeck coefficient of around $-480 \text{ } \mu\text{V/K}$. This in-plane Seebeck coefficient value was higher than its out-of-plane counterpart, which has only half the carrier concentration compared with in-plane direction (*Sci. Rep.* 2017, 7:8914). Similarly, the in-plane Seebeck coefficient of 2D material Bi₂Te₃ at 575 K is approximately $-90 \text{ } \mu\text{V/K}$, nearly 9 times greater than its out-of-plane counterpart, despite Bi₂Te₃ having

a significantly higher carrier concentration in the in-plane direction (*RSC Adv.* 2019, 9:14422).

Our graphene superlattice, characterized by partially overlapped square nanopores, induces electronic reconstruction that leads to periodically formed electron domain walls in the in-plane direction. These walls enforce Fermi electron confinement and an equipotential Fermi surface, causing significant alterations in the density of states, electron transport, and their energy-dependent values, thereby amplifying the Seebeck coefficient. In contrast, in the out-of-plane direction, only the carbon atoms that overlap experience van der Waals forces, leaving the exposed regions lacking such forces. This unique structural arrangement introduces a significant level of complexity to the carrier transition process between layers via quantum tunneling, requiring the overcoming of substantial energy barriers. Consequently, this mechanism imparts to the electronic structure in the out-of-plane direction characteristics similar to those of an electrical insulator, characterized by a limited density of states near the Fermi level and insensitive energy dependencies in electrical conductivity, along with a significant energy gap. Due to these factors, the Seebeck coefficient decreases in the out-of-plane direction, primarily attributed to the lack of carriers capable of overcoming the barrier, a critical requirement for generating a significant thermoelectric voltage in response to temperature gradients.

To address the reviewer's comment, we have added the following text to the revised Supplementary Information:

“We noted that the in-plane Seebeck coefficient of the graphene superlattice exceeds that in the out-of-plane direction, by a factor of approximately 8.8 to 6.3 in the temperature range of 300–500 K. This difference is attributed to the higher density of partially overlapped nanopores between graphene layers, facilitating the formation of periodic electron domains along the in-plane direction. In contrast, in the out-of-plane direction, a larger energy barrier, with insulator-like characteristics, is present, leading to a weaker Seebeck effect.”

REVIEWERS' COMMENTS

Reviewer #2 (Remarks to the Author):

The authors have addressed the questions.